# Spatial deconvolution of HER2-positive breast cancer delineates tumor-associated cell type interactions

Alma Andersson[1,8], Ludvig Larsson [1,8], Linnea Stenbeck [1,8], Fredrik Salmén[1,2], Anna Ehinger [3,4], Sunny Z. Wu [5,6], Ghamdan Al-Eryani [5,6], Daniel Roden [5,6], Alex Swarbrick [5,6], Åke Borg[4], Jonas Frisén[7], Camilla Engblom [7] & Joakim Lundeberg [1✉]

In the past decades, transcriptomic studies have revolutionized cancer treatment and diagnosis. However, tumor sequencing strategies typically result in loss of spatial information, critical to understand cell interactions and their functional relevance. To address this, we investigate spatial gene expression in HER2-positive breast tumors using Spatial Transcriptomics technology. We show that expression-based clustering enables data-driven tumor annotation and assessment of intra- and interpatient heterogeneity; from which we discover shared gene signatures for immune and tumor processes. By integration with single cell data, we spatially map tumor-associated cell types to find tertiary lymphoid-like structures, and a type I interferon response overlapping with regions of T-cell and macrophage subset colocalization. We construct a predictive model to infer presence of tertiary lymphoid-like structures, applicable across tissue types and technical platforms. Taken together, we combine different data modalities to define a high resolution map of cellular interactions in tumors and provide tools generalizing across tissues and diseases.

[1] Science for Life Laboratory, Division of Gene Technology, KTH Royal Institute of Technology, Stockholm, Sweden. [2] Hubrecht Institute-KNAW (Royal Netherlands Academy of Arts and Sciences) and University Medical Center Utrecht, Cancer Genomics Netherlands, Utrecht, the Netherlands. [3] Department of Genetics and Pathology, Laboratory Medicine Region Skåne, Lund, Sweden. [4] Department of Clinical Sciences Lund, Division of Oncology, Lund University, Lund, Sweden. [5] The Kinghorn Cancer Centre, Garvan Institute of Medical Research, Sydney, Australia. [6] St Vincent's Clinical School, Faculty of Medicine, Sydney, Australia. [7] Department of Cell and Molecular Biology, Karolinska Institutet, Stockholm, Sweden. [8] These authors contributed equally: Alma Andersson, Ludvig Larsson, Linnea Stenbeck. ✉email: joakim.lundeberg@scilifelab.se

Transcriptomic and genetic studies have revolutionized our understanding of cancer and led to the development of new diagnostic and therapeutic tools. Recent advances in single-cell RNA sequencing (scRNA-seq) technologies have provided extensive insight into the cellular composition of tumors[1–3]. Building on decades of clinical and pre-clinical research, numerous scRNA-seq studies have analyzed tumor-associated cells, some of which mainly focused on cancer cells[4–12], immune cells[13–19], or fibroblasts[20–22]. In these studies, cell types were split into finer distinctions based on their molecular profiles. Together, these cellular subsets form complex ecosystems that continuously interact with each other to govern tumor progression and treatment outcome; yet many fundamental mechanisms regarding how and where tumor-associated cell subsets interact remain unresolved.

Although scRNA-seq offers a high-throughput analysis of cells' transcriptomes, their spatial context is lost during tissue processing. In contrast, techniques such as immunohistochemistry (IHC) and in situ hybridization provide high spatial resolution, but often require a priori selection of targets, making these methods less suited for high-throughput exploratory analyses. In addition, recent work has shown that most tumor-associated cell states are complex and not easily defined by a few marker genes or surface receptors[2], which presents both a technical and financial challenge to targeted spatial methods[23]. Therefore, Spatial Transcriptomics (ST), developed by Ståhl and Salmén et al.[24], which provides spatially resolved and transcriptome-wide expression information, is highly suitable to investigate cell interactions and spatial aspects of gene expression in the tumor stroma.

Breast cancer remains an important public health concern, claiming more than a hundred lives every day in the US alone, and its often arduous treatment takes a significant toll on patients[25]. Breast cancer is divided into several subtypes, including HER2-positive tumors, which are defined by an enrichment of the HER2 (human epidermal growth factor receptor 2) expression by tumor cells[26,27]. An estimated 15–20% of all breast cancers tumors are HER2-positive, and these tumors usually exhibit aggressive growth and demand intense treatment[28,29]. Research into the molecular underpinnings governing breast tumor progression has yielded considerable clinical benefits for HER2-positive breast cancer patients; most notably, the introduction of HER2-targeted therapies that has drastically improved disease outcome[30]. Despite this progress, many patients with metastatic HER2-positive breast tumors still succumb to the disease due to primary or acquired resistance to anti-HER2 therapies[31]. Alternative strategies to treat HER2 cancer patients are therefore needed.

In the past decade, drugs targeting the immune system have increased patient survival in several different cancers[32]. Traditionally, breast cancer has not been considered an immunogenic disease, but an increased presence of tumor-infiltrating lymphocytes has consistently been associated with a favorable breast cancer prognosis[33]. Elevated tumor-associated lymphocyte levels are observed in triple-negative (TNBC) and HER2-positive breast cancer, suggesting that they are more immunogenic compared to other breast cancer subtypes[34]. Furthermore, tertiary lymphoid structures (TLSs), which are lymph-node-like structures that can form ectopically in tissues such as tumors, holds certain predictive power of treatment outcome in HER2-positive tumors[35–37]. However, to date, immune checkpoint blockade treatment has only been clinically approved for TNBC. There is therefore a need to understand fundamental mechanisms that define HER2-positive tumors' molecular profile and cell composition.

In this study, we seek to deepen our understanding of the spatial aspects of tumor-associated cellular relationships by applying ST to human breast cancer. More specifically, we use ST to survey spatial-gene expression and cell types in 36 samples collected from eight HER2-positive individuals. We examine intra- and interpatient heterogeneity using a number of different methods, including expression-based clustering and single-cell data integration. We here define: (i) shared spatial expression signatures among HER2-positive patients, (ii) high-resolution cell state colocalization patterns that expose a type I interferon-associated macrophage T-cell interaction, and (iii) a method to identify putative tertiary lymphoid-like structures in ST data.

## Results
HER2-positive tumors from eight individuals (patient A-H) were subjected to ST with three (adjacent) alternatively six (evenly spaced) sections obtained from each tumor ($n = 36$ sections) (Methods and Supplementary Figure 1). For brevity, we will refer to sections originating from the same individual as replicates. Figure 1 provides an overview of the analysis workflow and methods.

**Manual annotation and initial data characterization**. One section from each tumor was examined and annotated by a pathologist (A.E.) based on the morphology of the associated HE-image (Hematoxylin and Eosin). Regions were labeled as either: in situ cancer, invasive cancer, adipose tissue, immune infiltrate, or connective tissue (Fig. 2A and Supplementary Figure 2).

To explore the spatial-gene expression data, we first applied common techniques of normalization, and visualized it in 2D-space using UMAP (Methods and Supplementary Figure 3)[38]. Spatial capture locations (hereafter, spots) from different patients separated into isolated clusters, with a high degree of intermixing between each patient's replicates. This clustering pattern implies the presence of interpatient heterogeneity, as can be expected when working with tumor samples[39]. In scRNA-seq data, immune cells, but not tumor cells, typically mix between patients[16]. In ST, multiple, often different, cell types contribute to each spot's transcription profile (~0–200 cells/spot); therefore, even immune-rich spots tend to separate in a patient-specific manner[40]. Thus, to properly capture the nuances of each patient's molecular profile and not risk quenching weak signals, we initially analyzed each patient separately.

**Expression-based clustering**. For the patient-wise analysis, we split the data into mutually exclusive patient subsets, each being independently normalized with the intention to remove technical noise and batch effects. Next, we clustered our data using Seurat's shared nearest neighbor approach (clusters are referred to as pXcY; $X =$ patient id, $Y =$ cluster id)[41]. Spots neighboring in physical space were often assigned to the same cluster, and the clusters aligned with the pathologist's annotations (Fig. 2A–C and Supplementary Figure 2). Furthermore, the spatial arrangement of clusters and the fraction of spots in each cluster were consistent across replicates (Fig. 2D, Supplementary Figure 3–5).

**Cluster annotation**. To better understand what biological entities the expression-based clusters represented, we performed differential gene expression analysis, resulting in a set of marker genes upregulated within and characteristic of each cluster (Fig. 2F). The marker genes were selected by a combined cutoff with respect to their adjusted $p$ value and fold change (Methods and Supplementary Data 1). To assess enrichment of biological pathways, we queried the clusters' marker genes against the GO:BP (GO Biological Processes) database using g:Profiler[42]. Based on the set of marker genes and their associated pathways, we annotated the expression-based clusters (Supplementary Data 2); these

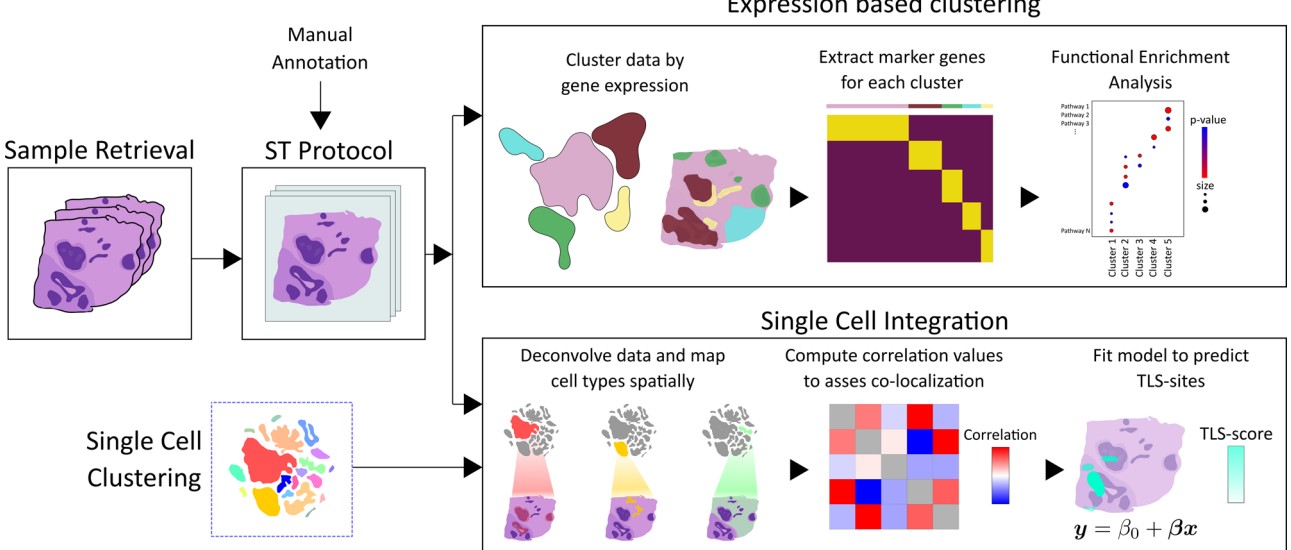

**Fig. 1 Overview of study.** After sample retrieval, we performed ST (Spatial Transcriptomics) on 36 sections confirmed to be HER2-positive. A trained pathologist manually annotated one section from each sample. Expression-based clustering and single-cell data integration were applied to explore the spatial expression profiles and cell type interactions in our data. Marker genes were extracted for each of the clusters and subjected to functional enrichment analysis, which allowed us to biologically annotate them. By deconvolving the expression profiles in each spot with the single-cell data, we could infer patterns of cell state colocalization and design a model for the prediction of tertiary lymphoid-like (TL-like) structures. The single-cell data and its associated cell annotations originate from an external source also examining HER2-positive tumor samples.

annotations are not exhaustive but serve to provide guidance for subsequent analysis.

**Comparison of clusters with pathologist-annotated regions.** We related the pathologist-annotated regions to the expression-based clusters by computing the fraction of spots within a region that belonged to each cluster (Fig. 2E). Strong concordance was observed; clusters enriched for immune-related processes overlapped well with the immune infiltrates and clusters with cancer-associated pathways fell into the cancerous regions (Supplementary Data 3). This comparison established an agreement between the pathologist's annotations and the clusters defined by our data-driven approach. Notably, an in situ cancer region that consisted of only three spots correctly clustered with physically separated, but identically annotated spots (pGc4 in Fig. 2B and Supplementary Figure 6), attesting to the sensitivity of ST and verifying that clusters are not only driven by physical proximity. In patient H, parts of a region labeled as in situ cancer were consistent, across all replicates, inhabited by two expression-based clusters; one (pHc1) that overlapped with the other cancer areas while the other (pHc4) was enriched for immune processes and aligned spatially with the annotated immune infiltrates (Supplementary Figure 6). These observations suggest that data-driven expression-based clustering captures signals that may be overlooked by visual inspection; thus in some cases providing a more in-depth and nuanced depiction of the tissue.

**Exploring intra- and interpatient heterogeneity.** Multiple tumor profiles may exist among a group of individuals, but also within a given patient. Characterization of this heterogeneity could aid in the design of more nuanced and personalized treatment regimens[43]. Therefore, we used the expression-based clusters as a framework to examine intra- and interpatient heterogeneity in our data.

Interestingly, we observed intrapatient heterogeneity at the transcriptome level in most of our patients; all patients except two (patient B and H) had more than one cluster labeled as cancer. Given the HER2-receptor's prominent role in the etiology of the

HER2-subtype, we also examined the *ERBB2* (encoding the HER2-receptor) expression across the cancer clusters. We observed significant differences in *ERBB2* gene expression (two-sided Mann–Whitney $U$ test, $p_{adj} < 0.05$) between clusters in the same patient, attesting to a certain spatial heterogeneity of *ERBB2* expression (Supplementary Figure 7 and Supplementary Table 1). Patient E exhibited varying transcription profiles in two spatially separated tumor foci assigned to different clusters (pEc3 and pEc4) (Supplementary Figure 2). While such observations could be a consequence of "overclustering", the two clusters were separated and non-neighboring in UMAP-space, which suggested distinct expression profiles. Both clusters had *ERBB2* listed as a marker gene and were enriched for pathways associated with cell growth, but one of the clusters (pEc3) displayed a high degree of enrichment for immune response-related processes (e.g., humoral immune response, $p_{adj} = 4.0 \times 10^{-5}$; immune system process, $p_{adj} = 8.0 \times 10^{-5}$) while apoptotic and regulatory pathways were enriched in the other (pEc4: e.g., cell death, $p_{adj} = 6.5 \times 10^{-9}$; apoptotic processes, $p_{adj} = 1.6 \times 10^{-8}$) (Supplementary Data 4). These findings implied that one tumor focus (pEc3) likely had a higher degree of infiltrating immune cells than the other. Thus, ST enables us to further spatially demarcate regions with distinct molecular profiles that have similar morphology.

**Immune and tumor core signatures.** To assess the presence of universal features in our data, we compared clusters across patients with respect to their marker genes, selected as described above. We reasoned that if two clusters shared a large number of marker genes, they should be considered more similar than if the converse was true. To translate this notion of similarity into a quantitative metric, we computed the Jaccard Index for every combination of cluster pairs. Five distinct supergroups (a.k.a, "clusters of clusters") emerged after the clusters were hierarchically grouped based on similarity (Methods and Supplementary Figure 8). Only supergroups with at least one shared gene among the members were considered robust, three of the five supergroups fulfilled this criterion. In the remaining three

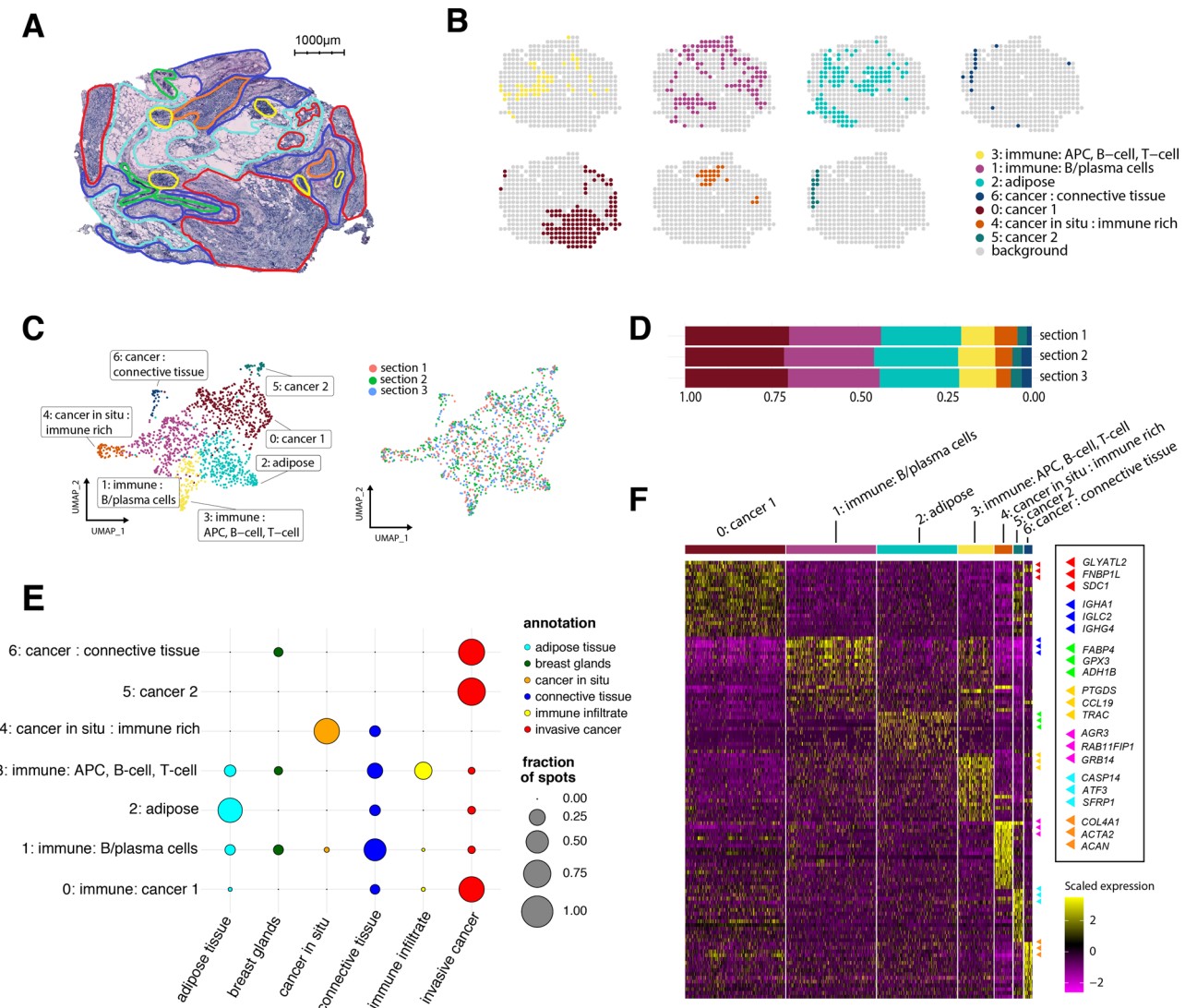

**Fig. 2 Results from the expression-based analysis of patient G. A** Morphological regions were annotated by a pathologist into six distinct categories: adipose tissue (cyan), breast glands (green), in situ cancer (orange), connective tissue (blue), immune infiltrate (yellow), and invasive cancer (red). **B** Split view of each expression-based cluster's distribution across one tissue section. **C** UMAP embedding of all spots ($n = 446$) from the three replicates, markers are colored based on cluster identity. **D** Proportions of spots assigned to each cluster across the three replicates. **E** Dot plot showing the overlap between clusters and annotated regions. The size of the dots represents the proportion of cluster spots belonging to an annotated region. The pathologist's annotations are given on the $x$ axis, cluster annotations on the $y$ axis. **F**. Heatmap of the clusters and the most highly differentially expressed genes. Each cluster was annotated based on its association with morphological regions and marker genes. Heatmap colors represent normalized and scaled expression values. APC stands for antigen-presenting cell.

supergroups, marker genes present in a majority of the member clusters (at least 80%) were considered representative core signatures. Two core signatures were immune-related: the first being a set of 47 genes, including *APOE* and *C1Q{A,B,C}* highly expressed by macrophages (Mø), suggesting that clusters in the corresponding supergroup might contain tumor-associated Mø[16]; the second was a set of 55 genes with lymphocyte and MHC class I–II-associated members (e.g., *TRBC1*, *HLA-{A,B}*, *HLA-D{QB1,RA,RB1}*)[44]. The third core signature consisted of 11 genes, with several of them being related to cancer and proliferative growth (e.g., *ERBB2*, *EPCAM*, and *CDH1*)[45,46]. This cancer core signature was derived from a supergroup where all clusters were annotated as cancer-associated. For clarity, the aforementioned cancer supergroup consisted exclusively of cancer-associated clusters, but not all such clusters were members of this group. Complete core signatures are found in Supplementary Data 6.

**Inference of cell type organization by integration with single-cell data.** Given how the spatial arrangement and patterns of interaction between different cell types have implications for both disease progression and treatment, we wanted to chart each cell type's spatial distribution within the tissue. However, ST data do not provide single-cell resolution, i.e., each spot hosts several cells of potentially different types, meaning that spots ca not always be exclusively assigned to a single cell type. A common strategy to estimate cell type abundance at the capture locations is to integrate ST and single-cell/nuclei RNA-seq data in order to deconvolve the mixed observations of the former, effectively mapping cell types into spatial context. Such integrative methods have previously been successfully applied to spatial cancer data[47]. To deconvolve our ST data, we employed the stereoscope method, which for every spot in the spatial data estimates the proportion of cells that belongs to each cell type defined in a given single-cell data set, producing a $n_{spots} \times n_{cell\ types}$ matrix of proportion

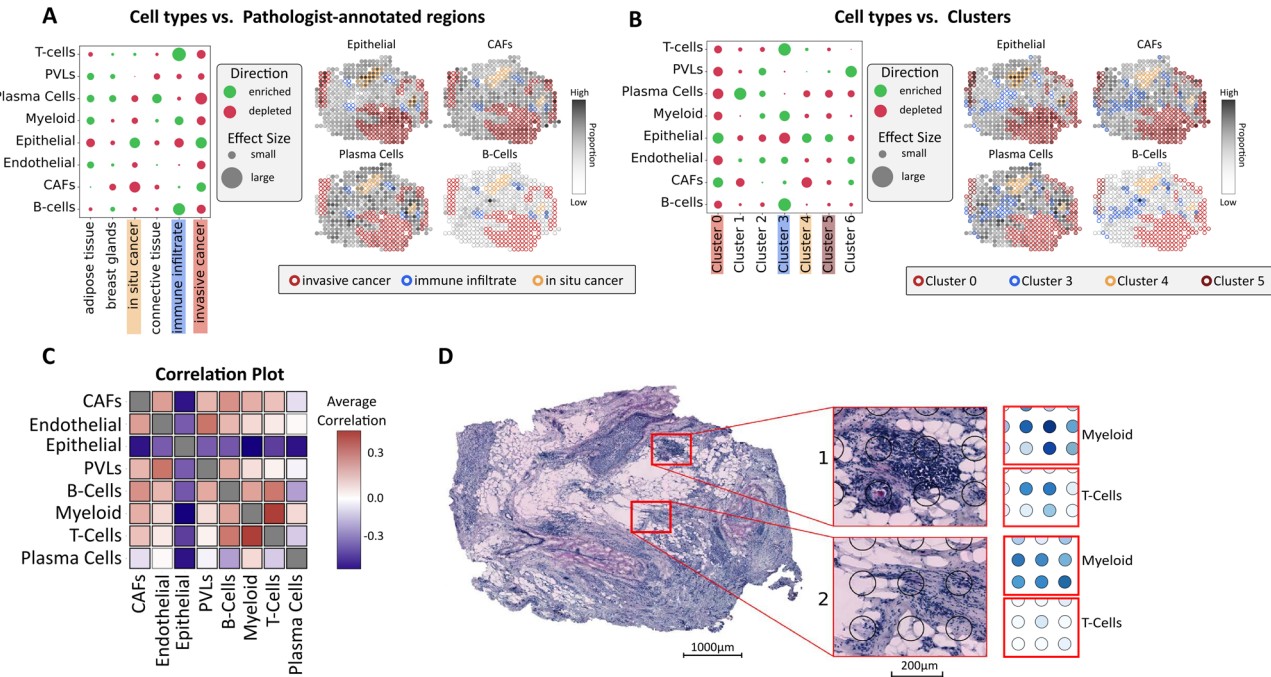

**Fig. 3 Spatial mapping of cell types from scRNA-seq data. A** Enrichment (green) and depletion (red) of major tier cell types in the regions defined by the pathologist (patient G), along with proportion estimates of different cell types (e.g., epithelial, CAFs, plasma cells, and B cells). Spot annotations are indicated by border colors, included regions are: in situ cancer (orange), invasive cancer (red) and immune infiltrate (blue). **B** Similar to **A** but with the regions defined by the expression-based clusters. **C** Correlation plot of all cell types within the major tier across all 36 sections and patients, a distinct correlation between myeloid cells and T cells can be observed. **D** Proportions of myeloid cells and T cells showing one area with higher (1), respectively, lower (2) degree of colocalization. All presented results are associated with patient G, except for subfigure C where correlation values are computed across all patients. *CAFs* cancer-associated fibroblasts, *PVLs* perivascular-like cells.

values[48]. The stereoscope method uses complete expression profiles, facilitating distinction between similar cell types with overlapping sets of marker genes, which is especially important in a complex and intermixed environment like tumors.

A scRNA-seq data set from five HER2-positive tumors, annotated in three tiers (Methods) was used to deconvolve our spatial data[49]. The three tiers are referred to as the major, minor, and subset tiers (terminology from Wu et al.). There were eight different cell types in the major class: myeloid cells, T cells, B cells, epithelial cells, plasma cells, endothelial cells, cancer-associated fibroblasts (CAFs), and Perivascular like cells (PVL cells). The minor tier represents finer partitioning of the major cell types, e.g., Mø and CD8+ T cells. In turn, the lowest tier further splits the minor tier into cell states or "subsets", such as chemokine-expressing Mø and IFNγ-expressing T cells. Excerpts from the integrative analysis are given in Fig. 3. Supplementary Data 7 contains a visualization of all of the remaining patients and tiers, all output from stereoscope is found in Supplementary Data 8.

**Interactive exploration of results**. We have compiled a resource that contains all data and results from the expression-based clustering and single-cell mapping; with a graphical user interface (GUI) that enables comprehensive exploration of these (Code Availability).

**Enrichment of cell types within manually defined regions**. We next examined cell type enrichment/depletion within the pathologist-annotated regions. Several affirmative trends were observed, B and T cells were enriched in the immune infiltrates while cancer regions showed enrichment of cancer-related cell types and depletion of several stromal cell types (patient G in Fig. 3A, all patients in

Supplementary Data 9). All patients except patient B showed enrichment of the HER2-related epithelial cancer type in invasive cancer regions (Supplementary Figure 9). In contrast, patient B exhibited depletion of the HER2-related epithelial cell type and enrichment of the luminal B (LumB) type in the invasive cancer regions. Coincidentally, patient B was also unique in having a progesterone-receptor positive profile, in concordance, with the LumB molecular subtype (Supplementary Figure 9 and Supplementary Table 2)[50]. Taken together, these findings were seen as affirmative of our mapping's validity.

**Enrichment of cell types within expression-based clusters**. We examined enrichment/depletion of cell types within the expression-based clusters to see how the single-cell mapping related to these (Fig. 3B and Supplementary Data 9). In patient E—with the two spatially disconnected tumor foci—the cluster associated with apoptotic and regulatory pathways (pEc4) was enriched for certain epithelial cells and depleted of memory B cells. In contrast, the immune-rich cancer cluster (pEc3) was enriched for memory B cells and CD4+ T cells, with weak or no enrichment of cancer types (Supplementary Figure 10). In patient G, all clusters annotated as cancer were: depleted of plasma cells, had very low enrichment, or were depleted of B and T cells, and three out of four clusters were enriched for epithelial types (Fig. 3B). The in situ cancer cluster (pGc4) was the only cancer cluster in patient G enriched for dendritic cells across all replicates while also being depleted of myofibroblast-like CAFs (Supplementary Figure 11). We also observed how the plasma cell immune cluster (pGc1) was enriched for plasma cells while the antigen-presenting cell (APC) immune cluster (pGc3) exhibited stronger enrichment of B cells, T cells, and myeloid cells. PVL cells were overrepresented in mixed cancer/connective tissue

clusters (pGc6), which also showed enrichment of myeloid cells, CAFs, and endothelial cells (Fig. 3B and Supplementary Figure 12). Agreement with the tissue morphology, pathologist annotations, and expression-based clusters suggest that the single-cell data are sufficiently representative of our tissues to provide a reliable mapping of the included cell types.

**Spatial maps of population diversity**. To obtain an overview of the cell population's diversity within different spatial regions, we computed the entropy for the cell type distribution (proportion estimates) within each spot (Methods). A high entropy score indicates high diversity while a low entropy score indicates the opposite. We found that the immune-related clusters were more heterogeneous than the cancer clusters, an effect more pronounced in patient A. The APC-enriched clusters, when present, tended to exhibit the highest degree of diversity, with clusters of patient B being an exception (Supplementary Figure 13).

**Trends of cell type colocalization**. The cell mapping was used to explore putative cellular interactions by computing the cell type spot-wise Pearson correlation (Fig. 3C), with a positive correlation between two cell types being considered as them colocalizing. At the major tier, the most prominent feature—present in all patients—was an anticorrelation between epithelial cells and every other cell type (Fig. 3C). The endothelial cells (major tier) also exhibited well-preserved colocalization patterns with CAFs (all patients except G) and perivascular cells (all patients except F) (Fig. 3C). At the minor tier, epithelial cells are split into cancer and normal epithelial cells, which anticorrelate in all patients except patient E (Supplementary Figure 14). For patient-wise colocalization matrices, we refer to Supplementary Data 10. The increased cell type resolution thus revealed how the cancer epithelial cells are the main contributors to the trend of epithelial cell anticorrelation observed in the major tier. There is also a spatial separation between the two CAF types at the minor tier (all patients except patient A, Supplementary Figure 14). At the subset level, mature Luminal cells (a subset of normal epithelial cells) always colocalize with one cancer type (except in patient A), which could suggest proximity to luminal cells or a mature luminal phenotype (Supplementary Figure 15). In the immune cell population, plasma cells are anticorrelated with B cells in all patients (Fig. 3B, C) except one (patient A, Supplementary Figure 16). These findings indicate that B cells and plasma cells, reside at distinct locations within the tumor[51]. It is not clear whether these findings reflect plasma cell migration during local differentiation from tumor-associated B cells, or if the plasma cells have developed from B cells outside the tumor microenvironment. Colocalization of varying strength between B and T cells (major tier) was observed in five of the eight patients; patients G and H exhibited particularly strong colocalization signals and their B and T-cell distributions had an ample overlap as shown below and in Supplementary Figures 17 and 18.

We also observed that T cells colocalized with myeloid cells (Fig. 3C, D); of interest, since interactions between T cells and myeloid cells are well established and can profoundly affect their respective behavior[52]. Recent studies have also revealed a substantial heterogeneity within T-cell and myeloid cell types, where their respective subsets exhibited a diverse spectrum of states[13,16,18]. When the finer tiers of T cells and myeloid cells were examined, several trends of colocalization could be observed; such as weak positive signals between cDC2:CD1C, Mø1:EGR1, and pDC:IRF7 with several CD4+ T-cell populations, including Tfh and Tregs. We also observed a salient correlation between Mø2:CXCL10 and a T-cell state (T cells:IFIT1) across all patients (Fig. 4A) and wanted to explore this further.

**Presence of type I interferon response processes**. As indicated in the scRNA-seq resource, Mø2:CXCL10 expressed increased levels of the chemoattractants *CXCL9* and *CXCL10*. Both of these chemokines bind CXCR3, typically found on T cells and NK cells[53,54]. Tumor-associated myeloid cells expressing *CXCL9/10* have previously been described and attributed important immunotherapy-induced antitumor functions[16,55–57]. Furthermore, *CXCL9/10* expression may be induced by type I interferon stimuli[49]. Similarly, several of the marker genes for the T-cell:IFIT1 state are also associated with a type I interferon response.

Type I interferon activation within tumors can act directly on tumor cells, to inhibit proliferation or stimulate apoptotic processes, or indirectly by activation of antitumor immunity[58,59]. In addition, certain anticancer therapies have been shown to induce and depend on type I interferon activation[59]. Given the relevance of type I interferon responses in cancer treatment, we wanted to evaluate its association with the Mø2:CXCL10 and T-cell:IFIT1 cell state colocalization signal in our spatial data. Thus, we inspected the cell "type within cluster" enrichment results and noted that a majority of the patients had at least one cluster (e.g., pGc4) enriched for both Mø2:CXCL10 and T cells:IFIT1 (Fig. 4B and Supplementary Data 9). Consequently, we aimed to make a more quantitative estimate of how this joint presence of the Mø and T-cell states was distributed among our clusters. For this purpose, we devised a joint score based on the cell type proportions, where high values indicate high proportion values of both cell types. Stratifying the spots from each sample by their cluster identity, several clusters with elevated signals of the joint score emerged, e.g., pBc3, pDc5, pEc1, and pGc4 (Supplementary Figure 19-Supplementary Figure 20), in agreement with the cluster-cell type enrichment results (Fig. 4B and Supplementary Data 9). Furthermore, interferon response-related pathways were enriched within all clusters with high joint scores (Supplementary Data 4).

**Spatial enrichment of type I interferon responses**. To chart the interferon pathway expression, we conducted a spatial enrichment analysis. Regions with high enrichment of the type I interferon pathways spatially aligned with areas of joint Mø2:CXCL10 and T-cell:IFIT1 presence (Fig. 4C–E and Supplementary Figure 21). Notably, the relationship between the type I interferon signal and the joint presence of the cell states appears to be asymmetric; joint Mø2:CXCL10 and T-cell:IFIT1 presence overlaps with an elevated interferon signal but the opposite is not always true. Such asymmetry is expected since other cell types are known to exhibit and be stimulated by interferon signaling pathways[59]. Still, our results imply that joint localization Mø2:CXCL10 and T-cell:IFIT1 subsets often occur in the presence of a type I interferon signal. We confirmed these findings in an independent HER2-positive spatial-gene expression data set, produced using the Visium platform (Supplementary Figure 22). Finally, we showed the presence of the same signal in a data set from a vastly different cancer form (squamous cell carcinoma, SCC) generated with two distinct platforms, ST2K and Visium (Supplementary Figure 23 and Supplementary Note 1)[60]. Our results suggest that a spatially restricted type I interferon response may relate to Mø-induced recruitment of particular T-cell subsets. Further investigations would be useful to establish whether these interactions are relevant to disease outcomes.

**Inferring tertiary lymphoid-like structures from cell type proportions**. Next, we returned to the patterns of B- and T-cell colocalization, more specifically how this related to TLSs. Our interest in TLSs stems from their cardinal role in antitumor

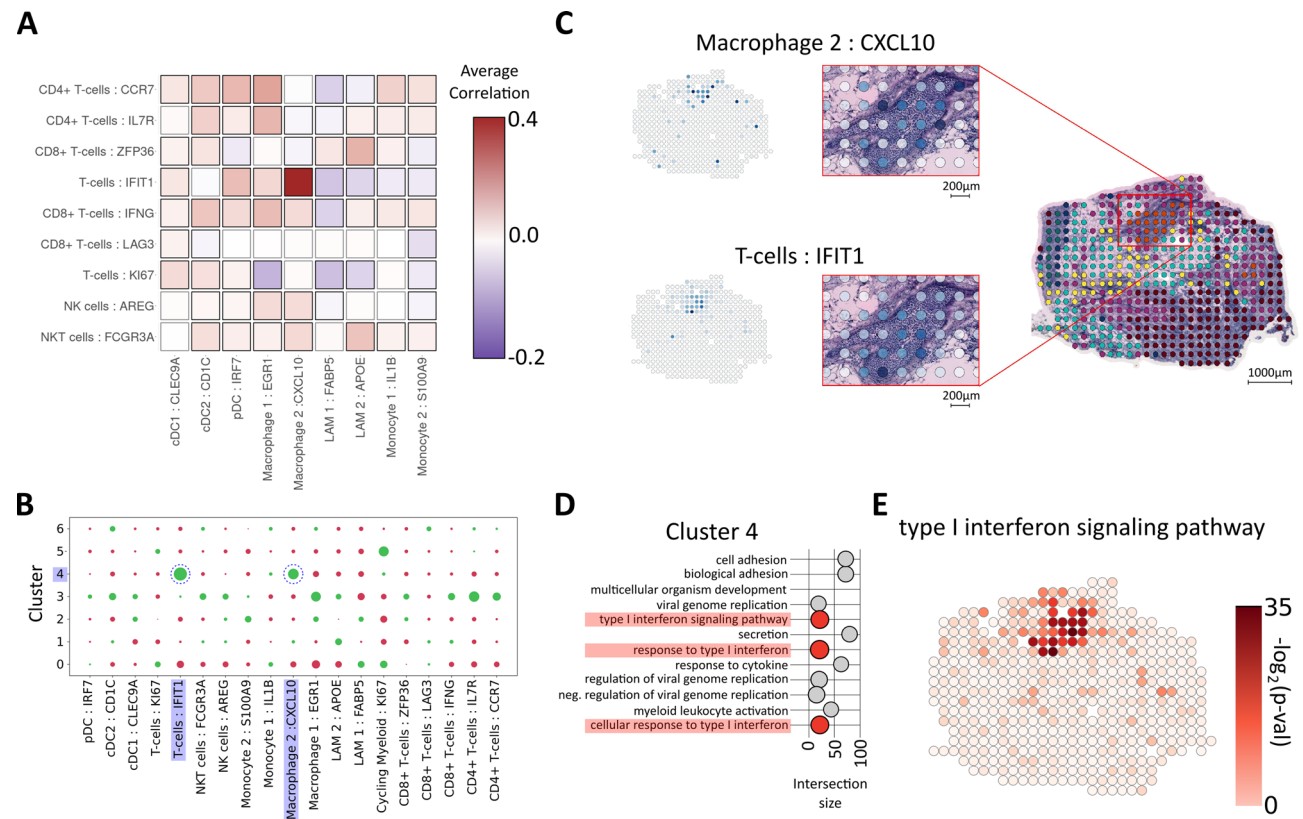

**Fig. 4 Colocalization of myeloid cells and T cells. A** Correlation plot of T- and myeloid cell subsets showing a distinct correlation between the T-cell:IFIT1 and Mø2:CXCL10 (macrophage 2:CXCL10) subsets across all patients. **B** Enrichment (green) and depletion (red) of subsets of T- and myeloid cells in each expression-based cluster, highlighting the presence of the correlated cell types T-cell:IFIT1 subset and Mø2 within cluster 4 of patient G (pGc4). **C** Proportion estimates for T-cell:IFIT1 subset and Mø2 in pGc4. **D** Pathways enriched by marker genes for pGc4, type I interferon signaling pathways are highlighted in red. Intersection size is equivalent to the number of overlapping terms between pGc4 marker genes and the given pathway. **E** Spot-wise enrichment of a type I interferon signaling pathway (GO:0060337) visualized on patient G, the *p* values which the enrichment scores are based on were computed using a two-sided Fisher's exact test (Methods), no adjustment for multiple hypothesis testing was applied since a single hypothesis is tested in each spot (enriched or not enriched).

immune responses and relation to the clinical outcome as well as treatment response. In the context of cancer, TLSs are one of the locations where tumor antigens are presented to T cells, promoting a more-targeted antitumor attack [61].

As TLSs are defined by the presence and interaction of multiple cell types, scRNA-seq techniques are suboptimal for studying them unless the sites are separated from the remaining tissue prior to dissociation. We, therefore, see our use of ST, where each spot represents a small neighborhood populated by multiple cells, as complementary to scRNA-seq methods when studying these structures. Although TLSs are not exclusively inhabited by B and T cells, they are implicated by their joint presence[62]. Having deconvolved the cell type composition of each spot, we were able to identify spots that exhibited a high degree of colocalization between B and T cells, ergo potentially constituting parts of a TLS-site that we will call TL-like structures. More explicitly we did this in a fashion similar to how the scores for the joint presence of Mø2:CXCL10 and T-cell:IFIT1 subsets were computed, resulting in a metric we refer to as TLS score (Methods). A positive TLS score translates to B and T cells both being enriched for at a site, negative values the opposite. As expected from the overlap in B and T-cell distribution, patient G and H exhibited small compartmentalized regions with high TLS scores (Fig. 5A) here considered as TL-like structures.

**Experimental validation of TL-like structures**. To validate that our estimates of joint B- and T-cell presence were suitable as

proxies for TL-like structures, we used IHC. We stained against the canonical cell type markers CD20 (B cells) and CD3 (T cells) on three additional sections from patient H, one double staining and two single stains (Methods). These sections were not adjacent to the sections subjected to ST, but as proximal as technically feasible in order to preserve as much as possible of the major histological regions. Our results showed characteristic features of previously described immature TLSs—compartmentalized structures of B cells surrounded by T cells, but lacking distinct germinal centers—in regions of elevated TLS signals (Supplementary Figure 24) [61,63].

**Characterizing the gene expression profiles of TL-like structures**. Next, we wanted to assess how the gene expression related to the TLS score. For this purpose, we used a simple linear model to predict the TLS score of a spot, based on its (normalized) gene expression, and then extracted the genes with the largest coefficients, i.e., highest contribution to a positive score; we refer to this set of genes as a TLS signature. The number of signature members (171 genes, Fig. 5B, Supplementary Data 5) was determined by a threshold derived from the trained model (Methods). The three genes with the largest coefficient values were: *MS4A1* (a well-known B-cell associated gene, encoding the antigen CD20), *B2M* (encoding a protein that interacts with and stabilizes MHC I), and *TRBC2* (encoding a component of the T-cell receptor). Other signature members have previously been associated with TLSs e.g., *CXCL13, CXCR5, CCL19,* and *LTB,* the latter two which

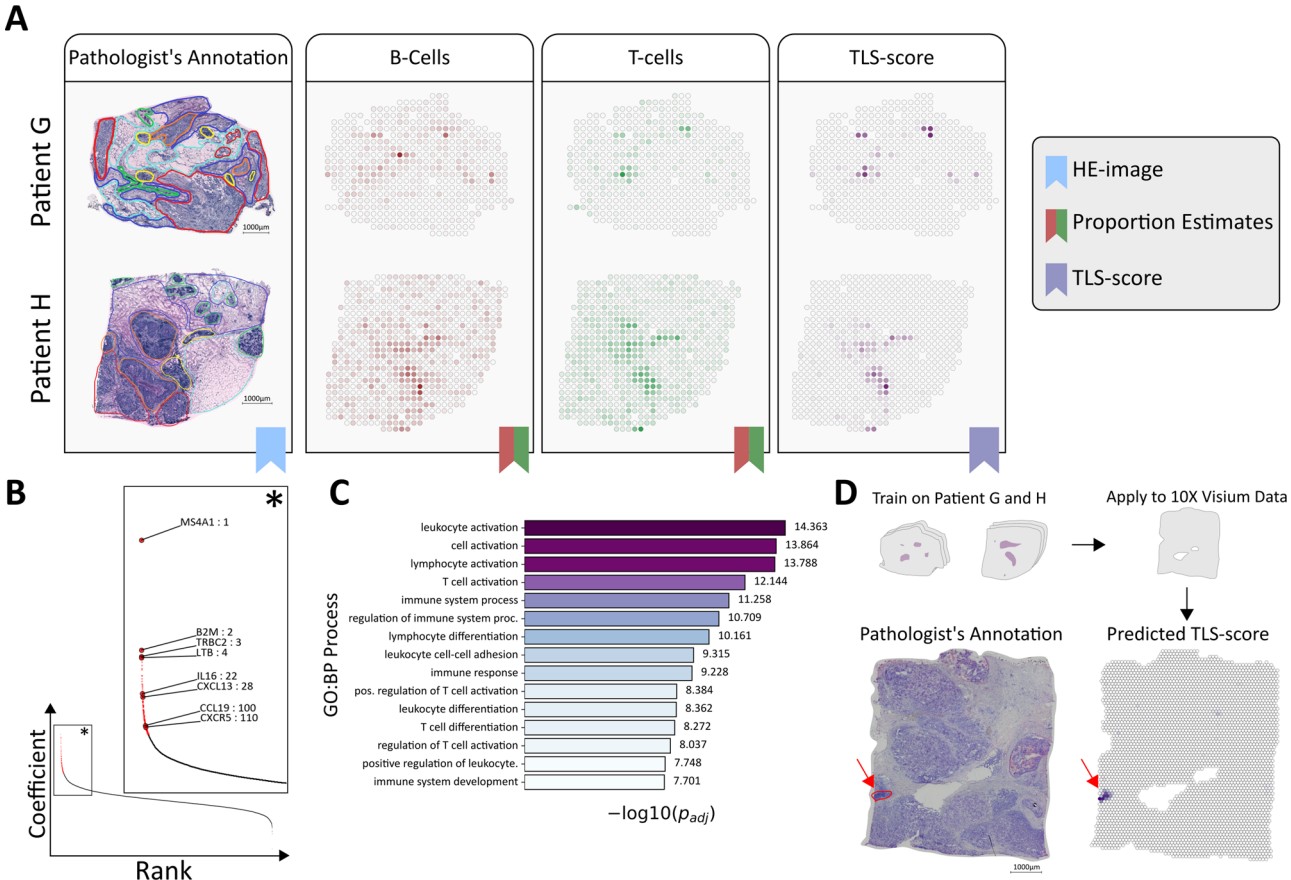

**Fig. 5 Inference and prediction of tertiary lymphoid-like Structures. A** Proportion estimate of B and T cells together with the computed TLS score (tertiary lymphoid structure) for patients G and H, annotated Hematoxylin and Eosin (HE) images are included for reference. **B** Rank-plot (coefficient value vs. rank) of the fitted model, genes included in the TLS signature are indicated by red; the zoom-in (*) shows the rank TLS-associated genes. **C** Top 15 pathways of which the TLS signature showed enrichment of, ranked according to the adjusted p value (as provided by g:Profiler). **D** Predicted TLS score for the 10x Genomics™ Visium breast cancer data set, using the model trained on patient G and H. Pathologist's annotation for likely TLS-sites (red) are included as a reference. Scale bars represent 1000 μm.

suggest the presence of other TLS-associated cell types in addition to B and T cells (Fig. 5B)[61]. To see what biological processes the TLS signature was enriched for, we subjected it to functional enrichment analysis (using g:Profiler, querying against GO:BP). The top processes were all related to cell activation, differentiation, and immune response or regulation (Fig. 5C and Supplementary Data 12).

**Model validation by cross-platform prediction and prediction of survival outcome.** To control for overfitting, we applied the model to (HER2-positive) breast cancer data originating from a different platform (Visium, downloaded from 10x Genomics™ website, see Data Availability). Strong concentrated signals were observed in the Visium data, overlapping with the region identified as a likely TLS-site by our pathologist (Fig. 5D). As an additional test, we applied our model to ST data from other tissue types: developmental heart, rheumatoid arthritis, and melanoma[64–66]. No TL-like structures were identified in the former whereas several TL-like structures were found in the latter two, as expected (Supplementary Figure 25 and Supplementary Note 2).

Although no clinical data were available for our patients, we reasoned that if the model we propose captures TLS-related features, it should be able to reproduce results from previous studies where such features are examined. We, therefore, applied our model to the same data set (TCGA skin cutaneous melanoma (SKCM) bulk RNA-seq) as Cabrita et al[67]. used to show how a

TLS signal's strength can be associated with overall survival. Encouragingly, we managed to successfully reproduce their result, linking patients of high predicted TLS score to better overall survival than patients with low or intermediate scores (Supplementary Figure 26 and Supplementary Note 3). Thus, our model not only generalized to other experimental platforms and external data but could also reproduce TLS-linked patient outcome results from bulk RNA-seq studies. These results strongly support our model's ability to identify TL-like structures.

## Discussion

We have used the ST technique to study the spatial-gene expression profiles of eight HER2-positive tumors. The work can be decomposed into two main parts.

First, our study followed an outline similar to many single-cell studies; we normalized and clustered the data, extracted marker genes for each cluster by differential gene expression analysis, subjected the marker genes to functional enrichment analysis, and finally used the entire corpus of information to annotate the clusters. From a joint analysis of the clusters, we derived representative core signatures of the shared tumor and immune populations in our data set. The expression-based clusters agreed with the high-level pathologist annotations, but also gave a finer partition of the tissues and information about each region's molecular profile. The value of this is manifold; manual annotation is labor-intensive, requires access to a trained pathologist,

and does not—like ST data—enable molecular profiling of the regions (e.g., pathways enrichment).

In the second part of our study, we used an integrative method (stereoscope) to spatially map cell types/states defined in a single-cell data set; which allowed us to survey the spatial distribution of the cell types and their patterns of colocalization. From the colocalization trends, we concluded that: the epithelial cancer types tend to be dominant when present; B cells seem to spatially segregate from plasma cells; there was a type I interferon-associated coupling between the certain T-cell and Mø states in some patients, and B and T cells colocalize in several patients. We particularly focused on two of these interactions, between T-cell/Mø subsets and B cells/T cells.

The proximal location between the *CXCL9/10*-expressing Mø and T-cell states, observed in several samples, suggests that Mø2:CXCL10 could be recruiting the IFIT1 expressing T cells into specific locations. We reproduced our findings—an interaction between Mø and T-cell subsets in the presence of a type I interferon response—in two external data sets: a breast cancer (HER2-positive) tissue section and a larger data set from a different cancer type (SCC), both data sets were profiled with different experimental platforms. Colocalization between Mø and T cells had already been shown in the SCC data set using antibody-based methods as well as expression of the receptor CXCR3 and its ligands CXCL9/10/11; the authors also report a type I interferon response being present in their data, but do not explicitly link the two signals together[60]. Although other publications have documented related findings, more extensive efforts are required to understand the mechanism of the trifold interaction and its role in the tumor ecosystem [13,14].

The other colocalization signal, between B and T cells, was used as a proxy for sites resembling TLSs (here, TL-like structures), an approach validated by IHC staining. The staining showed the expected B and T-cell compartmentalization, but could not confirm the presence of mature germinal centers, implying immature TLSs or variable tissue quality. We used a linear regression model to identify TL-like structures based on gene expression, and extracted a gene signature from the model's parameters. Several signature genes had previously been attributed important roles in TLS formation or function, and were not B- and T-cell specific, acting as further evidence of the method's legitimacy. Despite its simplicity, the model generalized across techniques and tissues, it also reproduced previous results linking TLS prevalence to clinical outcome. Although further studies are necessary to confirm our findings, transcriptional profiling of TL-like structures in this manner could potentially reveal therapeutic targets for drugs aiming to promote anticancer immune responses.

We chose the stereoscope method to map single-cell data since it does not rely on marker genes, rather it operates with expression profiles, which is beneficial when attempting to resolve cell states defined by few and not mutually exclusive marker genes. The mapped single-cell data are considered representative of our tissues but have some shortcomings. Importantly, cell types such as neutrophils, mast cells, and adipocytes were not present in the single-cell data. Other cell types (e.g., certain dendritic cell subsets) were too few in numbers to enable robust mapping and hence omitted from the mapping. Still, our ST data and previous studies suggest the presence of these cell types, which have been shown to have important tumor-associated functions, in the tumor tissues[2,68]. Absence of cell types could also lead to slight misrepresentations of the cell type distributions. Furthermore, there is a risk of transcriptionally dominant cell types masking less abundant ones, as a majority of the captured transcripts would originate from the former. Finally, we believe that cell subsets defined by a proliferative state could incorrectly be

mapped to the same region—i.e., presenting a false colocalization signal—if the proliferation process strongly dominates the expression profile of these cells, akin to observation in previous studies[69]. However, these concerns pertain to all methods relying on a reference data set, and are not unique to stereoscope. Taken together, we regard the integrative and expression-based clustering strategies as complementary methods that together establish a holistic image of our data.

Our study shows how ST data can be used to chart the molecular profiles of tissue samples in a disease context. It also hints at the potential impact future similar studies might have if coupled with clinical data. For example, cell colocalization patterns may be linked to patient outcome, used to assess drug responses in a spatially restricted manner within tumors, and to study functional interactions. In addition, the development of computational methods that enable the assessment of complex, non-linear, and hierarchical interactions between multiple cell types would benefit any prospective study.

To conclude, we demonstrate a transferable workflow for the comprehensive analysis of ST data applied to HER2-positive breast cancer tumors. Several findings emerged from our analysis that, if further explored, may help to better understand the underlying disease mechanisms and open up for new vantage points for therapy.

## Methods

**Array production**. The microarrays were generated as a $33 \times 35$ grid of printed spots with a 200 μm center-to-center distance of 100 μm between each capture location (spot). A total of 1007 spots were printed with unique DNA oligonucleotides (spatial barcodes) attached to oligo(dT) capture probes [24].

**Sample acquisition**. After surgery, the tumors used for this study were trimmed of fat and divided up into several pieces, immediately frozen at −80 °C, and then stored in a tumor bank until the start of the experiment. For each tumor, sections from a different piece of the tumor—to that used in the ST experiments—were subjected to IHC and PAM50 analysis for the classification of subtypes. All analyzed sections from the tumors chosen for this study were stained positive for HER2 and were classified as HER2-positive tumors by PAM50. The PAM50 analysis was performed prior to the initialization of this study and was only used for sample selection. The samples were collected in concordance with the Declaration of Helsinki and the study was approved by the Regional Ethical Review Board of Lund before initiation of the work (Dnr. 2009/659). All patients were provided with full verbal and written information about the study before their participation. Written informed consent was obtained from all patients.

**Tissue handling, staining, and imaging**. The protocols used in our study have previously been described in Ståhl et al.[24,70] and a detailed version of the entire protocol is available in Nature Protocols. In short, fresh frozen material was sectioned at 16 μm. After placing the tissue on top of the barcoded microarray, the glass slide was warmed at 37 °C for 1 min for tissue attachment and fixed in ~4% formaldehyde (Sigma-Aldrich, F8775) for 10 min at room temperature (RT). The slide was then washed briefly with 1× PBS (phosphate-buffered saline, Medicago, 09-9400). The tissue was dried with isopropanol (Fisher Scientific, A461-4) before staining. The tissue was stained with Mayer's hematoxylin (Agilent, S3309) for 4 min, washed in Milli-Q water, incubated in bluing buffer (Agilent, CS702) for 2 min, washed in Milli-Q water, and further incubated for 1 min in 1:20 eosin solution (Sigma-Aldrich, HT110216) in Tris-buffer (pH 6). The tissue sections were dried for 5 min at 37 °C and then mounted with 85% glycerol (Merck, 104094) and a coverslip. Imaging was performed using the Metafer VSlide system at ×20 magnification. The images were processed with the VSlide software (v1.0.0). After the imaging was complete, the coverslip and remaining glycerol were removed by dipping the whole slide in Milli-Q water followed by a brief wash in 80% ethanol and warming for 1 min at 37 °C.

**Permeabilization and cDNA synthesis**. Using the pre-permeabilization mix for most tissue types from the protocol, the pre-permeabilization was carried out using 0.4% Collagenase 1 (Thermo Scientific, 17018-029) in BSA (Bionordika, B9000S) and HBSS buffer (Thermo Fisher Scientific, 14025-050) and was incubated for 20 min in 37 °C. The pre-permeabilization was followed by incubation in 0.1% pepsin-HCl (Sigma-Aldrich, P7000-25G, pH 1) for 10 min at 37 °C to permeabilize the tissue. A cDNA-mix containing Superscript III (Thermo Fisher, 18080085), RNaseOUT (Thermo Fisher, 10777019), 0.1 M DTT (Thermo Fisher, included with Superscript III), dNTPs (Thermo Fisher, R0191), BSA (Bionordika, B9000S), and

Actinomycin D (Sigma-Aldrich, A1410-2MG) was added and the slide incubated at 42 °C overnight (~18 h). The tissue was washed with 0.1× SSC between each incubation step.

**Tissue removal and cDNA release from the surface**. For tissue removal the two-step protocol was used[70], in brief beta-Mercaptoethanol (Calbiochem, 444203) was diluted in RNeasy lysis buffer (Qiagen, 79216) and the slide was incubated for 1 h at 56 °C. The wells were washed with 0.1× SSC followed by incubation with proteinase K (Qiagen, 19131), diluted in proteinase K digestion buffer (Qiagen, 1034963), for 1 h at 56 °C. The wells were then washed in 2× SSC + 0.1% SDS followed by 0.2× SSC and lastly with 0.1× SSC and dried. The released mix consisting of second-strand buffer (Thermo Fisher, 10812014), dNTPs (Thermo Fisher, R0191), BSA (Bionordika, B9000S), and USER enzyme (Bionordika, M5505L) was added to each well and incubated for 2 h at 37 °C. After probe release, the 1007 spatial spots containing non-released DNA oligonucleotide fragments were detected by hybridization of fluorescently labeled probes (Supplementary Table 3) and imaging, in order to obtain Cy3-images for image alignment and spot detection, as described in the protocol [24,70].

**Library preparation and sequencing**. No modifications were made to the protocol, but parts of it were carried out using an automated pipetting system (MBS Magnatrix Workstation), also previously reported[70,71]. In general, second-strand synthesis and blunting were carried out by adding DNA polymerase I (Thermo Fisher, 18010025), RNase H (Thermo Fisher, 18021071), and T4 DNA polymerase (Bionordika, M0203L). The libraries were purified and amplified RNA (aRNA) was generated by a 14 h in vitro transcription reaction using T7 RNA polymerase (Sigma-Aldrich, AM1334), supplemented with NTPs (Sigma-Aldrich, AM1334) and SUPERaseIN (Sigma-Aldrich, AM2694). The material was purified and an adapter-ligated to the 3′-end using truncated T4 RNA ligase 2 (Bionordika, M0242L). Generation of cDNA was carried out at 50 °C for 1 h by Superscript III (Thermo Fisher, 18080085) in first-strand buffer (Thermo Fisher, included with Superscript III), RNaseOUT (Thermo Fisher, 10777019), DTT (Thermo Fisher, included with Superscript III) and dNTPs (Thermo Fisher, R0191). Double-stranded cDNA was purified, and full Illumina sequencing adapters and indexes were added by PCR using 2xKAPA HotStart ready-mix (Roche, KK2602). The number of amplification cycles needed for each section was determined by qPCR with the addition of EVA Green (Biotium, 31000). Final libraries were purified and validated using an Agilent Bioanalyzer and Qubit before sequencing on the NextSeq500 (v2) at a depth of ~100 million paired-end reads per tissue section. The forward read contained 31 nucleotides and the reverse read 46 nucleotides.

**Mapping, gene counting, and demultiplexing**. The forward read contained the spatial barcode and a semi-randomized UMI sequence (WSNNWSNNV, with W—A/T, S—G/C, N—A/C/T/G, and V—A/C/G) while the reverse read contained the transcript information and was used for mapping to the reference GRCh38 human genome. Before mapping the reads with STAR[72], the reverse reads were first quality trimmed based on the Burrows–Wheeler aligner, long homopolymer stretches were also removed. Multi-mapped reads i.e., read mapping to multiple loci in the genome, were discarded after mapping with STAR. HTSeq-count with the setting -intersection-nonempty, was used to generate gene counts, using an Ensembl reference file (v. 86)[73]. The remaining reads were provided as input to TagGD demultiplexing using the 18 nucleotides spatial barcode[74]. The demultiplexed reads were then filtered for amplification duplicates using the UMI with a minimal hamming distance of 2. The UMI-filtered counts were used in the analysis. The analysis pipeline (1.6.0) is available at https://github.com/SpatialTranscriptomics Research/st_pipeline.

**Pre-processing**. Raw data was merged from six section gene count matrices for samples A, B, C, and D and three-section gene count matrices for samples E, F, G, and H. The merged expression matrices were enriched for genes matching the biotypes protein_coding, IG_C_gene, IG_J_gene, IG_V_gene, TR_C_gene, TR_J_gene, and TR_V_gene. In addition, each merged expression matrix was filtered from ribosomal protein genes (RPL and RPS), mitochondrial genes (MT-), and MTRNR genes as well as genes expressed in fewer than 10 spots across the whole merged data set. Spots with fewer than 300 unique features (genes) were also removed from the merged data sets.

**Normalization and feature selection**. The merged data were first normalized using the regularized negative binomial regression method implemented in the SCTransform function from Seurat (v3.1.4) R package[75]. The number of variable genes selected with SCTransform was determined by applying a residual variance cutoff of 1.1 (variable.features.rv.th = 1.1) with the additional parameter settings; return.only.var.genes = FALSE and variable.features.n = NULL. In the subsequent patient-based analysis, we applied the same normalization scheme but with an additional batch correction term to adjust for technical differences across replicate tissue sections (vars.to.regress = section).

**Dimensionality reduction**. Before running dimensionality reduction, the set of highly variable genes as defined by the SCTransform method was reduced to a smaller set of genes as described below. First, we hypothesized that the most relevant features should not only have high variance but also show positive spatial autocorrelation. We, therefore, devised a method to rank the variable features by spatial autocorrelation by computing the Pearson correlation coefficient for each gene between the expression vector and the spatial lag vector (defined as the summed expression in the adjacent neighboring spots over all spots). Variable genes with a correlation coefficient larger than 0.1 were therefore kept in the reduced gene set. We also identified 21 highly variable genes which contributed to form a ring-like pattern in several capture areas (Supplementary Data 11). This effect was not found in all biological replicates from the same tissue biopsies and was therefore concluded to be a source of technical variation. All 21 genes were excluded from the reduced gene set. For each patient data set, the reduced set of highly variable and spatially correlated genes was used as input for a Non-negative Matrix Factorization (NMF) computation with 10 factors using the RunNMF function from the STUtility R package[76]. Each factor was then visualized as a spatial heatmap colored by factor value and factors with consistent patterns across replicate tissue sections were kept for subsequent analysis steps.

**Expression-based clustering**. First, a Shared Nearest Neighbor (SNN) graph was constructed from the selected NMF factor matrix with the FindNeighbors function in Seurat. This SNN graph was then used to identify clusters of spots using the modularity-based clustering algorithm implemented in the FindClusters function in Seurat. The resolution parameter was set to 0.4 for all samples.

**Marker detection**. For each patient data set, a Wilcoxon signed-rank test was performed using the FindAllMarkers function in Seurat to find differentially expressed genes within each cluster. The function performs the test pairwise between each cluster and its background (all other spots in the data set). The resulting table of gene markers was filtered to include genes with an adjusted p value lower than 0.01 and an average log fold (natural logarithm) change higher than 0.15, thus omitting down-regulated genes.

**Cluster annotation**. Each set of differentially upregulated genes were subjected to enrichment analysis using the Gene Ontology—Biological Processes (GO:BP) database and the enricher function from the g:profiler R package with an adjusted p value cutoff of 0.05. Each cluster was then manually annotated using the top enriched pathways and upregulated marker genes as a basis.

**Cluster overlap between patients and core signature extraction**. To check for overlapping gene signatures between clusters from different patients, we computed the Jaccard index for all pairs of cluster gene sets (upregulated DE genes). These values were first used to compute a distance matrix (euclidean distance) from which a dendrogram was constructed (R package hclust) with the agglomeration method set to "complete". This dendrogram was then cut into five groups using the cutree function (R package stats) with k (number of clusters) set to 5. Then, for each group of clusters, we extracted all genes that were shared between at least two clusters. For two of the groups, zero genes were shared across all clusters and these groups were excluded. For the remaining three groups, we defined a core signature as the genes that were shared across at least 80% of the clusters.

**Single-cell data**. We downloaded the single-cell data related to Wu et al.[49] publication. Only cells originating from the HER2-positive patients were used in our analysis. We used the same labels as in the figures of the single-cell resource, with the exception of plasmablasts which we here refer to as plasma cells.

**Spatial mapping of single-cell data**. To infer the spatial organization of certain cell types we used a method developed to integrate spatial and single-cell data, implemented and available as a python package (stereoscope, v.0.2, https://github.com/almaan/stereoscope). The method is based on a probabilistic model, which assumes that both single-cell and spatial RNA-seq data follow a negative binomial distribution. By using annotated single-cell data in combination with ST data it estimates proportions of every cell type (present in the single-cell data) at each spatial capture location [77].

*Breast cancer single-cell data*. To conduct the spatial mapping of cell types, we only included cells from the five HER2-positive patients found in the single-cell data, all 36 ST-sections were used. In total three analyzes were conducted, with the only difference being the labels used for the single-cell data. As mentioned in the main text, every cell was assigned to a type within each of the three tiers *major, minor,* and *subset*. For respective tier, we subsampled the single-cell data set, according to the following scheme: (i) If a type had fewer than 25 members, excluding the type; (ii) if a type had >25 members but ≤500 members, includes all cells; (iii) if a type had >500 members, randomly select 500 of these. Next, the subsampled sets were spatially mapped, one by one, onto the ST data.

A custom gene list of 5540 members, representing the union of the 5000 (highest expressed) genes in the single-cell data and cell type marker genes, were

used for the proportion inference, see Supplementary Data 13. In all, 50,000 epochs and a batch size of 2048 were used for all tiers, in both steps of the stereoscope procedure. Default values were used for all remaining parameters.

The fitted parameters were used to map the same single-cell data set onto a Visium sample from 10x Genomics™ (see Data Availability); though we only mapped the subset tier. We used the same settings during the mapping (50,000 epochs and a batch size of 2048).

*Squamous cell carcinoma.* To validate the trifold interaction between T cells, Mø, and a type I interferon signal we used an additional data set, downloaded from the GEO repository (accession code: GSE144240). The GEO entry contains both ST and scRNA-seq data from SCC tissues, both used in our mapping. The ST data consisted of eight ST2K (ST second generation, 2000 spots/array) sections and four Visium samples, and the scRNA-seq data 48,164 cells with associated metadata.

When using stereoscope we applied the same subsampling strategy as for the breast cancer data set, using a lower bound of 25 cells and an upper bound of 500 cells. For both steps in stereoscope 50000 epochs were used together with a batch size of 2048, all genes were included in this analysis (i.e., no top selection or custom gene list).

**Cell-type colocalization.** We use spot-wise Pearson correlation between the estimated cell-type proportion values as a proxy for cell type colocalization; with high positive correlation being indicative of types that exhibit similar spatial distributions and the opposite being true for negative values. The Pearson correlations were computed across all spots for each pair of cell types; more specifically, if we let $p_{sz}$ denote the proportion of cell type $v$ at spot $s$ the Pearson correlation $r_{uv}$ between two cell types $u$ and $v$ is given in Eq. 1:

$$r_{uv} = \frac{\sum_{s \in S}(p_{su} - \bar{p}_u)(p_{sv} - \bar{p}_v)}{\sqrt{\sum_{s \in S}(p_{su} - \bar{p}_u)^2}\sqrt{\sum_{s \in S}(p_{sv} - \bar{p}_v)^2}} \quad (1)$$

where $S$ is the set of all spots and $\bar{p}_i$ represents the mean proportion value across all spots for cell type $i$, see Eq. 2:

$$\bar{p}_i = \frac{1}{|S|}\sum_{s \in S}p_{si} \quad (2)$$

To estimate the confidence interval for each of the correlation values, we used a bootstrap approach. The Pearson correlation was computed for each of 10,000 bootstrap samples (sampling from all spots with replacement), forming a distribution of correlation values for each pair of types. The mean of each distribution was taken as a representative correlation value, and a 95% confidence interval was defined by the 2.5th and 97.5th percentiles. Pairs, where the confidence interval included zero, were considered as not statistically significant, indicated with a gray border.

**Entropy as a measure of cell-type diversity.** A region might be considered homogeneous if it's predominantly inhabited by one or a few cell types, while a heterogeneous—or highly diverse—area would represent a mixture of multiple cell types with similar abundance. With this interpretation of heterogeneity, the concept of entropy (as defined in information theory) can be used to gauge how diverse a certain spot is with respect to the cell type population. The entropy of a spot ($E_s$) with an associated proportion vector $\boldsymbol{p_s}$ is in Eq. 3:

$$E_s = -\sum_{z \in Z}p_{sz}\log_2(p_{sz}) \quad (3)$$

where Z is the set of all cell types and $p_{sz}$ is the proportion of cell type $z$ in spot $s$. Maximal entropy would be observed for those spots with uniform, very heterogeneous, cell-type composition while spots where a single-cell type dominates have low entropy values [78].

**Region-based enrichment/depletion of cell types.** The enrichment, or alternatively depletion, of the mapped cell types in relation to spatial regions (e.g., manual annotations or clusters) were assessed by the following procedure: First, the average proportion value was computed within each of the regions, referred to as the *true average.* Next, we permuted the spot indices for the proportion estimate vectors 10,000 times, while maintaining the original indices for the annotated regions. In other words, the proportion estimates were shuffled w.r.t. their spatial location. Average proportion values of the annotated regions were determined for each permutation, constituting the set of *permuted averages.* We then computed the differences between the true average and all permuted averages. Finally, the mean value of the differences divided by the standard deviation of these differences was taken as the *enrichment score* for respective regions.

Upon visualization, the *enrichment score* of a type within a certain region is represented by two features, the marker size, and its color. We let the marker size be proportional to the absolute value of the enrichment score, while the color indicates the sign (red for negative and green for positive). To summarize, red markers are indicative of a type being depleted in a certain region, green markers of enrichment; the larger the marker, the larger the effect.

**Joint score for Mø and T-cell subset interactions.** To examine the prevalence and spread of the presumed interaction between the Mø2:CXCL10 and T-cell:IFIT1 subsets we employed a strategy to compute a joint score based on their estimated (by *stereoscope*) proportion values. For every sample, we multiplied the paired proportion values for each subset in each spot, and then applied a z-transformation (subtraction by the mean followed by division with the standard deviation). High positive signals indicate the increased joint presence, low negative signals are the opposite.

**TLS signature.** The method we devised to obtain a TLS signature can be decomposed into two steps: (1) associating a TLS score to each spatial location and (2) modeling the contribution of each gene to this score. We base the TLS score on the proportion values estimated for the major cell type tier; first, raw TLS scores are computed, taken as the product between B- and T-cell proportions multiplied by the scalar 2. In theory—assuming unbiased and independent sampling from a large population of cells with the same type composition as the spot—this represents the probability that a pair of two randomly selected cells consists of a B and T-cell. The raw TLS score is then adjusted by subtracting the average probability of picking any cell type pair in the associated spot, this is the final TLS score used in the subsequent steps.

In the second step, we consider the (adjusted) TLS score at a given spot as a function of its gene expression. The gene expression values are normalized accordingly: First, all elements of a spot's expression vector are divided by its library size (the sum of all elements); second, the expression vector associated with each gene is divided by its standard deviation (computed after the preceding library size division).

Let $\mathbf{y_s}$ represent the TLS score vector for spots $s$ and $x_s$ the normalized expression vector of spot $s$. Using ordinary least squares, we then estimated the coefficients ($\beta_0$, $\boldsymbol{\beta}$) of the linear model $\mathbf{y} = \beta_0 + \boldsymbol{\beta}^T\mathbf{x}$. Implementation-wise, we used the *OLS class* from the *linear_model* module in the python package statsmodels (version 0.11.0), no regularization terms were used.

The genes qualifying as members in the final TLS signature were determined by ordering the coefficient values from largest to smallest, considering the values as a function (*f*) of the gene's rank. The resulting curve is then smoothed with a gaussian filter, and the second-order differences of this smoothed curve are computed, representing an approximation of *f*'s second derivative. Using the same gaussian filter as previously mentioned, the second derivative approximation is smoothed. The gene coefficient for which the smoothed second derivative approximation obtains a value below zero is taken as the lower bound (threshold), hence all genes with a coefficient having a lower rank than this will be excluded. The Gaussian filtering was performed by using the *gaussian_filter* function from scipy's (version 1.4.1) *ndimage* module; the sigma parameter was set to 10, while default values were used for the remaining parameters. Applying the aforementioned procedure to all replicates of patient G and H, we obtained a signature of 171 genes, full list in Supplementary Data 5.

Functional enrichment of the gene signature was performed by using the *g:Profiler* python package (version 1.0.0), we queried against GO:BP (GO Biological Processes) and selected all terms that were significantly enriched (having an adjusted $p$ value smaller than 0.05). The complete set of these pathways are found in Supplementary Data 12.

**Spot-wise pathway enrichment.** To assess enrichment of a given gene set (here associated with a given functional pathway) spatially, we used the following approach. Let G be all genes present in the spatial data, and let $Q_p$ be the set of all genes associated with the pathway P for which enrichment should be examined. Also, for each gene subtract its average expression, then within each spot rank the genes according to their adjusted expression levels (from highest to lowest). Let $T_n(s)$ be the set of the $n$ highest-ranked genes within spot $s$. Now, for each spot construct a $2 \times 2$ contingency table illustrated in Table 1:

To then conduct a Fisher's exact test (two-sided), in order to calculate the probability ($p(s)$) of observing this partitioning of genes among the two variables, assuming that the genes associated with P are equally distributed over the top ($T_n(s)$) and lower-ranked genes. The enrichment score ($E_p(s)$) of P for spot $s$ is presented in Eq. 4:

$$E_p(s) = -\log_2[p(s)] \quad (4)$$

These are the values visualized in Fig. 4E.

**Table 1 Contingency table explaining the setup for the Fisher's exact test conducted in order to find spot-wise enriched genes.**

| | Top ranked genes | Low ranked genes |
|---|---|---|
| Associated with P | $|T_n(s) \cap Q_p|$ | $|Q_p \setminus T_n(s)|$ |
| Not associated with P | $|T_n(s) \setminus Q_p|$ | $|G \setminus (T_n(s) \cup Q_p)|$ |

**IHC validation**. The tissue was sectioned at 10 μm and placed onto microscope slides (Thermo Fisher Scientific, J1800AMNZ) with two sections placed on each slide. In total, four sections were taken have one for the negative control, one for each of the single stains, and one for the dual staining. After sectioning, the slides were stored at −80 °C overnight.

The slides were then thawed at RT until condensation disappeared and then fixated for 10 min in 100% acetone (Sigma-Aldrich, 179124) that was pre-chilled to −20 °C. After fixation, the slides were dried and a hydrophobic barrier was drawn (Vector Laboratories Inc, H-4000) around each of the sections and then the sections were rehydrated in PBS for 5 min. All the following incubations were performed in a humidity tray and slide rack (ACD HybEZ™, PN310012, and PN321716) to maintain humidity during incubations and in RT. The sections were incubated in BLOXALL (Vector Laboratories, SP-6000) for 15 min and then washed three times for 2 min in PBS. Next, the sections were incubated in ImmPRESS 2.5% normal horse blocking serum (Vector Laboratories, S-2012-50) for 45 min, the serum was blotted off using paper (VWR EU, 115-0202) to then incubate in the primary antibodies for 1 h. The sections were incubated either with anti-CD20 antibody (1:50 mouse monoclonal, Abcam tab9475), anti-CD3 antibody (1:50 rabbit polyclonal, Dako A0452), an equal mix of the two, or just PBS (negative control). After incubation, the slides were washed three times for 5 min in PBS and then incubated for 30 min with ImmPRESS Duet HRP/AP Polymer Reagent (Vector Laboratories, MP-7724-15) after which they were washed three times for 10 min in PBS.

IHC signal was developed using ImmPACT DAB substrate (Vector Laboratories, SK-4105) for 3 mins and then washed in water for five minutes and rinsed once in PBS before developing the second signal using ImmPACT Vector Red substrate (Vector Laboratories, SK-5105) for 2.5 mins and then the slides were washed three times for two min in PBS. The sections were then counterstained using Gill's Hematoxylin (Sigma-Aldrich, GHS132) for two min and then rinsed in water 10 times after which the slides were dried and then mounted using ProLong Antifade Gold (Thermo Fisher Scientific, P36930) and stored in +4 °C overnight. Images were acquired using Nikon Eclipse Ni-E upright motorized microscope at ×4 and ×20 magnification.

**Validation of antibodies for IHC**. The chosen antibodies were first validated on tonsil tissue, which has clear lymphoid follicles and is optimal to test for B- and T-cell staining (results not included). Four sections were utilized for the validation step (10 μm thickness): one section for dual staining, one per antibody for single staining (two in total), and one as a negative control. The protocol was then optimized on another breast cancer tissue to ensure compatibility with the specific tissue type. The same setup as for the tonsil tissue was used. The final IHC validation was performed once on the actual tissue due to the nature of the tissue, being that the morphology is changing for each section and it being critical to stain for B- and T cells as close as possible to the section used for ST.

**Prediction of clinical outcome based on TLS score**. To show generalizability and clinical relevance of the model we devised to predict TLS score and thus infer the presence of TL-like structures, we sought to reproduce the findings of Cabrita et al.[67], where they relate levels of TLS to survival outcome among patients with metastatic melanoma in the TCGA database.

We accessed the TCGA data by using the two R packages *RTCGA.clinical* and *RTCGA.rnaseq* (version 1.60.0), from which we extracted the SKCM RNA expression and clinical data. We filtered the tumors to only keep those classified as metastatic based on the information in the *patient.clinical_cqf.tumor_type* column of the *SKCM.clinical* object. Survival data was extracted by using the *survivalTCGA* function, only samples where survival data were available were kept. The RNA-seq bulk expression data were extracted using the *expressionTCGA* function and normalized, within each spot division by library size followed by division of gene standard deviation within each gene. Next, we applied our linear TLS model to the normalized values and stratified the predicted score into quantiles. We then let the first, third, and fifth quantiles represent "low", "intermediate", and "high" groups of TLS signals. Finally, we generated Kaplan–Meier plots with the three strata and assessed whether differences in trends existed between the three groups; this was done using the *survfit* and *Surv* function from *survival* package and *ggsurvplot* from the *survminer* package.

**Reporting summary**. Further information on research design is available in the Nature Research Reporting Summary linked to this article.

## Data availability
The raw sequencing files for the ST data generated in this study are available with restricted access at the European Genome-Phenome Archive (EGA) under the identifier EGAD00001008031, access can be obtained by contacting Åke Borg (ake.borg@med.lu.se). The processed count matrices derived from the raw ST data and the associated brightfield images (HE-images) are available at https://doi.org/10.5281/zenodo.4751624 . The public data for the Visium breast cancer sample can be accessed at 10x Genomics's support website (https://support.10xgenomics.com/spatial-gene-expression/datasets, sample: Space Ranger 1.1.0, Human Breast Cancer (Block A Section 1)). The public data sets of the developmental heart (time point: 6.5PCW, sample: 4), melanoma (sample id: ST_mel1_rep1), and

rheumatoid arthritis (sample id: RA_B_3) can all be accessed via www.spatialresearch.org. The public spatial and single-cell data of Human SCC can be accessed at the Gene Expression Omnibus (GEO) under the accession code GSE144240. The public processed single-cell data from Wu et al. can be accessed via the Broad Institute Single-Cell portal at https://singlecell.broadinstitute.org/single_cell/study/SCP1039. Public raw sequencing data from the Wu et al. study is accessible at EGA under the identifier EGAS00001005173 and access can be obtained by contacting the Data Access Committee. SKCM data from the TCGA database is publicly available and can be accessed from the two R packages *RTCGA.clinical* and *RTCGA.rnaseq* (versions 1.60.0). The remaining data are available within the Article or Supplementary Information.

## Code availability
All code, data, and results that relate to the content of this manuscript are accessible via the GitHub repository https://github.com/almaan/her2st. The corresponding DOI is as follows: https://doi.org/10.5281/zenodo.5511762[79]. The repository also includes the results produced in the analysis and the code used for this purpose. Parts of the presented results (clustering and single-cell integration) can be interactively explored through a Shiny app interface, which is found at the aforementioned GitHub repository, further instructions are also provided in this location.

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

## Acknowledgements

We want to thank Patrik Ståhl for his valuable comments and advice throughout the process of this study. Furthermore, Jari Häkkinen and Johan Vallon-Christersson provided feedback and comments during the initial phases of this project, which we appreciate tremendously. We would also like to thank Mathew Tata for valuable help and guidance with the IHC staining experiments, and Fredrik Pontén for insightful comments. Finally, we thank Kim Thrane for sharing her melanoma data with us. This work was supported by the Knut and Alice Wallenberg Foundation, Swedish Cancer Society, Swedish Foundation for Strategic Research, the Swedish Research Council, Tobias Stiftelsen, Torsten Söderbergs Foundation, the European Union's Horizon 2020 research and innovation program under the Marie Sklodowska-Curie grant agreement no. 844712 (CE) and Science for Life Laboratory. We also thank the National Genomics Infrastructure (NGI), Sweden for providing infrastructural support. This work is further

supported by a research grant from The National Breast Cancer Foundation (NBCF) of Australia (IIRS-19-106) and the Petre Foundation. A.S. is a Senior Research Fellow of the National Health and Medical Research Council of Australia.

## Author contributions

A.A. and C.E. wrote the manuscript with input from the remaining authors. L.S. carried out the laboratory experiments and wrote the corresponding Methods part. A.A., C.E., and L.L. performed data analysis, A.E. (referred to as the pathologist) inspected the patient samples and performed morphological annotation. Å.B. provided samples and assessment of the biological findings. J.L., J.F., F.S., and Å.B. planned the ST study. S.W., G.A.E., D.R, and A.S. provided early access to the single-cell data used in the study as well as guidance regarding how to orient interpret the results related to it. All authors read and approved the manuscript.

## Funding

## Competing interests

A.A., L.L., L.S., C.E, J.F., and J.L. are scientific consultants for 10x Genomics Inc., providing spatially barcoded slides. The remaining authors declare no competing interests.
