## [Peer Review File · Nature Communications]

Spatial deconvolution of HER2-positive Breast cancer delineates tumor-associated cell type interactionsREVIEWER COMMENTS

Reviewer #1, expert in single cell sequencing (Remarks to the Author):

Using spatial transcriptomics, the authors assessed spatial gene expression in eight HER2-positive breast tumors, reporting on shared expression signatures among patients as well as intra- and inter-patient heterogeneity. By integrating spatial and single cell RNA sequencing data, they examined the spatial relationship between different cell types and found cell type co-localization patterns. Further, they built a model to predict tertiary lymphoid structures (TLS).

This is a well-written and interesting paper and one of the first to try to bridge spatial transcriptomics with single cell dissociative techniques to elucidate disease biology. While the sample size and lack of outcome data preclude the investigation of the relationship between cell topography and clinical features, a number of insights are nonetheless enabled by the integrated analyses described within. Overall, the authors present a thorough analysis and interpretation of these data and I commend them on generating a GUI to enable further exploration of the data. I have several questions and comments regarding the analysis.

Major points:

-I am curious as to how the ST signal is impacted when fewer tissues sections are combined – for example 3 rather than 6. There would appear to be a tradeoff in terms of the extent of heterogeneity captured when combining sections since this will span multiple cells. An analysis of this nature would help to explain the observed patterns of heterogeneity.

-Would it be possible to couple cell segmentation on adjacent H&E sections as has been done for slide-seq to improve the resolution on cell-cell interactions and their spatial context? This might afford an additional opportunity to integrate with single cell RNAseq.

-The authors observed inter- and intra-patient heterogeneity of tumor foci. While heterogeneity at both levels might be expected, can the authors elaborate on the presumed contributors to intra-patient heterogeneity between tumor foci? For example, in patient E there are two distinct tumor clusters, one cluster had a higher degree of infiltrating immune cells than the other. Are there any intrinsic factors associated with the higher degree of immune infiltrates?

-The TLS-signature indicate the co-occurrence of B and T cells in a spot. Since a ST spot corresponds to a small region of a tumor, what is the application of the TLS-signature to bulk RNAseq profiling of tumors.

Minor points:

-Some aspects of the results could be moved into the Methods section – for e.g. “Manual annotation”, “Initial data characterization”

-Text labels on Figure 2C is hard to read.

Reviewer #2, expert in breast cancer molecular pathology (Remarks to the Author):

- The paper entitled Spatial Deconvolution of HER2-positive Breast Tumors Reveals Novel Intercellular Relationships contains solid data on spatial transcriptomics of Her2+ breast cancers. It makes a first important step to better understand the organization and cell interaction in tumors and which will be very valuable for the community. However, on the formal side, I believe the authors could have added more details to allow the reader to understand and better navigate in the data. The study is decidedly interesting and clearly provides the first Spatial data in breast cancer sections but often not so well explained. It felt like it makes the reader dig into the material to understand it.

- Another important formal hurdle is that the authors use single cell RNA-seq data from an unpublished paper to deconvolve their spots and see which cell types are found in each spot. This important half of the data is missing.

-

Below a list of comments questions which I hope may help to improve the manuscript.

- The introduction is superficial and ill-suited to the content of the paper. It stresses on the clinical need for Her2 patients, but one could say that the authors have chosen this subtype because we understand to higher degree its origin and treatment: it is characterized by single major driver (the HER2 receptor), which is easily visualized and in many cases successfully targeted by antibody (trastuzumab and like), which for many works well. It is the primary and acquired resistance that render studies like the one presented here. In addition, the introduction could have engaged the curiosity of the reader by summarizing what we know about the types of cells in a breast tumor from scRNA seq, which is already a lot, to motivate on why it is important to also know where these cells are in space.

- It is stated that the pathologist annotation is an important first step to validate the results, i.e. to show a correspondence between the visual inspection and the expression data. While the plot in Fig 1E shows how the pathologist and ST agree, it could be interesting to understand more the disagreement, f.ex. to test whether happens at the margins or more globally.

- It is unclear whether the initial Her2 staining has been used by the pathologist to identify the cancer cells or not. Or if it used at any moment in the manuscript for example when two cancer 'clones' are identified at for a patient. Are these two equally Her2 positive?

- Additional validation based on IHC could be very valuable for example for the IFN section.

- It is unclear how the different sections from the same patient are used through the Manuscript? It seems that the sections end up showing very similar results? If true one needs a discussion on the heterogeneity and the analysis of different sections.

- It is almost surprising how clearly the patients are separating in the UMAP Supp Fig2. While I would clearly understand that the spots containing malignant cells may cluster by patients, I would expect that spots containing mainly immune cell would cluster together, in a patient independent way. Have the authors explored different normalization methods? Because I believe they would gain additional information by performing merged spot analysis instead of patient specific.

- How many cells are captured by spot in average? Can the clustering show doublets? Multiplets? I will come back to that down with the deconvolution.

- As the authors managed to deconvolve which cell types are present in each spot, is it possible to infer which cell type often 'dominates' spots when present in a mixture?

- It is unclear if stereoscope uses the single cell data to find which single cell RNA-seq cluster / cell type map to a spot or if it searches which different cell types may be found in each spot? I find the second more interesting, the first would merely be an extrapolation on what cell type dominate the spot?

- I believe the authors developed stereoscope because ST-clusters seemed general when merging all patients from the beginning. Would it be possible to identify more specific clusters when merging data from patient-specific cluster with similar annotation?

- The data used for the deconvolution of spots are not available and not even described yet, this a major drawback for this report. Stereoscope seems interesting but (i) is poorly described, (ii) is not testable, (iii) we have no clue of the data that have been used, (iv) we can't read the paper describing these data.

- It is surprising that there is so few cell types on the y-axis of Figure 3A and 3B while in the supplementary Figures there are much more cell types inferred from single cell data which seems to make much more sense.

- If stereoscope assigns the cell type dominating a spot and not all cell types found in a spot, knowing that there are several cells in a spot, would it be erroneous to seek which spots are found closer together (Fig3C) than what is expected by chance, would one need to search for which cells are more often found in the same spot than what expected by chance? The statistical analysis does not come through.

- Can the data be used to understand if local inhibitory signals in the tumor do not allow B cell to differentiate in plasma cells?

- The work on TLS is interesting and shows how powerful the ST can be, at his stage the authors mention they deconvolve the cell type composition of each spot, which I guess recall one of my previous comment above. if I understand it correctly the authors should be able to give some data on how many cell types are found in each spot and within clusters if spots may seem to be more

heterogeneous or not.

Minor

- In the beginning of the results, some Figures are shown before being cited example Figure 2B and 2C
- Please work on the legends of Figures, there are many clusters and colors and it is sometimes difficult to follow which cluster is mentioned and with each color. For example, legends are missing for Supp Fig.5.
- It is complicated to follow the terminology major / minor tier when one has not read the paper (in preparation) introducing this terminology.
-

Reviewer #3, expert in immune single cell sequencing (Remarks to the Author):

Andersson et al. used single-cell mRNA data to deconvolve spatial transcriptome and identified cell type differences in different spatial regions. They reported that expression-based clustering enables data-driven tumor annotation and assessment of intra-and interpatient heterogeneity. They discovered segregated epithelial cells, interactions between B and T-cells and myeloid cells, co-localization of macrophage and T-cell subsets by using Spatial Transcriptomics technology. They also constructed a model to infer the presence of tertiary lymphoid structures, applicable across tissue types and technical platforms. According to their finding, authors bring out new tools and biological insights into the spatial organization in HER2-positive breast cancer tumors, define novel interactions between tumor-infiltrating cells, and may open up for new therapy for HER-2 positive breast cancer tumors.

It is an interesting work with novelty, which combined tumor sequencing strategies with spatial information to explore HER-2 positive breast cancer. Even though the method is innovative, and the data is reproducible, there is not much controversy in terms of bioinformatics. Authors also need to depict more common characteristics or subpopulation analysis among all eight tumor samples instead of, for example, major attention to patient G and H. Co-localization of immune cells, such as TLS, CAFs, with cancer cells could be explored as well. Clinical significance of their finding were fully presented. This study does not seem to provide sufficient detailed insights into the role of B and T-cells and myeloid cells in tumor microenvironment with compelling translational implications.

Major points:

1. The pathologist in Figure 2B identifies cell types in different areas. But is there any counting of the number and proportion of different types of cells in different regions? Whether these ratios can be used as a reference for further deconvolution.
2. The UMAP plot labeled with different samples in Supplementary Figure 2A shows that the samples of different patients present a discrete distribution, which indicates that the transcriptome of each patient is different. However, if one has analyzed single-cell RNAseq, will know that only cancer cells (Malignant cells) show differences between patients, so how to explain the differences between patients in this entire region contains all types of cells?
3. The author proposed the stereoscope algorithm to deconvolve the spatial transcriptome. However, the window of deconvolution seems to have only selected 1 spot. Is it possible to try to deconvolve a window of 2x2 or 4x4 spots to test the robustness of the algorithm on a larger spatial scale?
4. Cell interactions need to connect with clinical significance.
5. Validate expression of the myeloid cell gene signature and the T-cells gene expression modules in the in TCGA.
6. More common characteristics of all eight tumor samples should be described such as cell types and co-localization information instead of, for example, major attention to patient G and H.
7. More cell subpopulation analysis should be presented. Besides interactions between B and T-

cells and myeloid cells, author should deconvolve cell state relationships in the tumor microenvironment.

8. Co-localization analysis of immune cells and cancer cells could be explored instead of co-localization analysis restricted only among immune cells.

9. The connection of TLS-score which was based on its (normalized) gene expression with clinical significance like disease outcomes could be added to further verify the function of TLS identified by the model. Because the real TLS include many other immune cells besides T and B cells, and the T and B cells might be in some specific states in TLS.

10. Type I interferon promoting macrophage-induced recruitment of T-cells could be further investigated its relationship with disease outcomes.

Minor points:

1. There was only one pathologist to estimate the various region. Two or more pathologists work together to judge might be more reliable.

2. Plots in Figures should be labeled with A, B, C., for example, Supplementary Figure 12-14

Reviewer #4, expert in single cell sequencing/spatial transcriptomics (Remarks to the Author):

This manuscript reports interesting new data and analyses, which will be of interest to cancer biologists. I have the following main concerns:

The paper is not transparent regarding the data and methods used. I do not believe that it would be acceptable to publish this manuscript before the data from the other unpublished paper is made available (ref. 30, used in throughout the manuscript). Second the authors use the stereoscope method (Figure 3, ref 22) but this has not been published. How can we evaluate the use of this method when it has not been published? For example, the method is similar in output to the Moncada et al publication (ref 21); however this is not acknowledged or compared.

In the section on spatial enrichment of type I interferon response the results are only shown for patient G. The authors should show a result that holds for more of the patients and show that in a main Figure.

The validation of the TLS signature on a single other instance in Fig. 5D is not sufficient. Also since the pathologist is part of the team I'm not sure that it can be seen as an independent annotation.

The statistics is generally not well documented in the manuscript. P-values should be given in the text , here for example: "but one of the cluster displayed a high degree of enrichment for immune response related processes (cluster 3, patient E) while apoptotic and regulatory pathways were enriched in the other (cluster 4, patient E), see Supplementary Data 6 ." Is the reader expected to go into the supplementary data file to find whether the result is statistically significant?

The writing in the Results section is not well-organized and contains many statements that are not fitting for it. For example, "Of course, the presence of multiple cancer clusters does not necessarily reflect different tumor types, but rather that the corresponding spatial regions are not homogenous in their expression; for example, as a consequence of hosting distinct immune cell populations. If additional metadata could be obtained for patients, ST might be useful for relating tumor intrapatient heterogeneity to more quantitative metrics, such as survival or treatment response."

HER2 Reviewer Responses

Reviewer 1

Using spatial transcriptomics, the authors assessed spatial gene expression in eight HER2-positive breast tumors, reporting on shared expression signatures among patients as well as intra- and inter-patient heterogeneity. By integrating spatial and single cell RNA sequencing data, they examined the spatial relationship between different cell types and found cell type co-localization patterns. Further, they built a model to predict tertiary lymphoid structures (TLS).

This is a well-written and interesting paper and one of the first to try to bridge spatial transcriptomics with single cell dissociative techniques to elucidate disease biology. While the sample size and lack of outcome data preclude the investigation of the relationship between cell topography and clinical features, a number of insights are nonetheless enabled by the integrated analyses described within. Overall, the authors present a thorough analysis and interpretation of these data and I commend them on generating a GUI to enable further exploration of the data. I have several questions and comments regarding the analysis.

R1 | Major Comment 1a

Comment : *I am curious as to how the ST signal is impacted when fewer tissues sections are combined – for example 3 rather than 6. There would appear to be a tradeoff in terms of the extent of heterogeneity captured when combining sections since this will span multiple cells. An analysis of this nature would help to explain the observed patterns of heterogeneity.*

Answer: To assess how the number of sections impacts the ST-signals, we selected one patient with 6 replicates (patient C) and reran our clustering analysis 6 times, each time increasing the number of included samples by one. That is, in the first iteration we only used one replicate, in the third iteration we used 3 replicates and in the sixth iteration, we used all replicates. We used the same clustering procedure as described in the main text (Methods), using the same “resolution” parameter in the SNN throughout all iterations. Spatial arrangement and UMAP embeddings for all six iterations are shown below in Review Figure 1.

Review Figure 1 | Incremental expression-based clustering analysis of patient C. The number of samples included in the clustering analysis increases from 1 to 6 (left to right). Top row shows spatial coordinates with spots colored according to cluster identity. Bottom row shows the UMAP embedding of the data in gene expression space, with spots colored according to cluster identity.

From these results, we are able to make two key observations:

- The number of sections or replicates impacts our ability to identify spatial structures in our data. With only one section there is poor separation of the data points belonging to different clusters in the UMAP embeddings, as more samples are added - the better separation of clusters we observe.
- The number of clusters (using the same resolution parameter) tend to increase with the number of samples as well, likely a consequence of better separation between spots.

Thus, heterogeneity is increased as more samples are included, expected as we cover more volume, meaning the likelihood of observing various molecular structures increases with the number of samples. However, previous studies have shown that gene expression and clusters are smooth in all spatial axes (see video at <https://www.molecularatlas.org/> and Review Figure 15) when studied with ST, hence we would attribute the changes we see when adding more samples to the fact that more data is available rather than biological diversity in the z-axis; to elaborate:

Using methods for dimensionality reduction (as is praxis before clustering), one assumes that the data resides on a manifold embedded in the high dimensional space, and seek to somehow characterize this manifold and the data's distribution within it. For example, data points in a three dimensional space might actually all be arranged along an one dimensional curve as can be seen in Review Figure 2, the low-dimensional embedding would here be the curve. In Review Figure 2A we see the **true curve** along which the data is arranged, easy to deduce from the large set of data points we may observe along it. However, Review Figure 2B represents what would happen if we sample the space sparsely and only collect a few observations; we are then prone to construct an incorrect manifold approximation and subsequently assign incorrect neighbors to some of the data points. In short, more data allows for better reconstruction of the data-manifold, which will produce more distinct clusters and enhances the downstream results. It is also in this context that one may speak of saturation when enough data to construct a sufficiently good manifold approximation has been gathered.

Review Figure 2 | A) True data distribution, three dimensional data is arranged in a one-dimensional curve embedded in the higher dimensional space. The dashed line represents the curve and the bottom the true order of data points. **B**) similar to A) but with missing data points (gray), an incorrect manifold approximation (dashed line) is made which leads to the data points exhibiting an incorrect order.

Revisiting our results from Review Figure 1, we see some changes in the spatial arrangement of the clusters after more than three sections are added, hence our dataset is not saturated, but the changes are fairly small. Also, for small clusters with few members, data from only one or two samples might not provide enough data to fully identify this as a standalone cluster.

Of course, more data is always preferable, but we are inclined to say that 3 replicates manage to capture much of the structures observed with 6 replicates and are sufficient to draw conclusions of a patient's molecular composition.

We hope this discussion shed some light on the reviewer's question. We have not incorporated these results in the main text, but could consider it if the editor or reviewer find it necessary.

R1 | Major Comment 2

Comment : *Would it be possible to couple cell segmentation on adjacent H&E sections as has been done for slide-seq to improve the resolution on cell-cell interactions and their spatial context? This might afford an additional opportunity to integrate with single cell RNAseq.*

Answer : Coupling of cell segmentation to the inference process could theoretically be incorporated into the model, for example as a way to introduce a lower bound for the proportion values, to exemplify: If 10 cells are located at a spot, no proportion value should be lower than 0.1 (1 / 10) since this would represent less than one cell. The - unpublished - method Tangram (<https://www.biorxiv.org/content/10.1101/2020.08.29.272831v1>) leverage such information in their optimization-based approach (though mainly for image based Spatial Transcriptomics techniques like MERFISH and seqFISH). The method we are using (now published in Communications Biology), *stereoscope*, however does not currently support incorporation of cell segmentation information.

stereoscope was specifically designed for sequencing-based methods (e.g. ST, Visium and Slide-seq), where only some platforms provide paired HE-images of the tissue. Therefore, we partially sought to develop a method that would not rely on segmentation data. Furthermore, performing cell segmentation on images of the fairly thick tissue sections used in ST and Visium, with multiple cell layers in the z-axis, is non-trivial and we are not aware of any robust methods devised for this purpose. Introduction of cell segmentation information would also introduce a lot of uncertainty in the inference process and potentially confound the results. We have tried

to apply cell segmentation software in prior studies, and noted how the aforementioned issues are very prominent in cell dense regions (which we have plenty of in our cancer data).

Nevertheless, we fully share the reviewer’s opinion that inclusion of additional data (such as cell segmentation) could enhance the *stereoscope* results and would be a valuable addition to the method itself. In future updates of the *stereoscope* software, we will most certainly keep this suggestion in mind.

R1 | Major Comment 3

Comment : *The authors observed inter- and intra-patient heterogeneity of tumor foci. While heterogeneity at both levels might be expected, can the authors elaborate on the presumed contributors to intra-patient heterogeneity between tumor foci? For example, in patient E there are two distinct tumor clusters, one cluster had a higher degree of infiltrating immune cells than the other. Are there any intrinsic factors associated with the higher degree of immune infiltrates?*

Answer : We understand and relate to the reviewer’s interest in better understanding the cause and mechanism of the intra-patient heterogeneity, it’s obvious how this might have implications in treatment design and drug development.

To address the question of whether any intrinsic factors of the tumor foci that could be related to the degree of immune infiltration existed, beyond different sets of marker genes and enriched pathways we looked at a more narrow set of genes, namely those found in the SRA (smallest region of amplification) in HER2-positive breast cancers, which among others include the genes: *ERBB2*, *GRB7*, *STARD3*, *PNMT*, *PPP1R1B* and *MLN64_[1,2]*. Here we could see that *GRB7* is coexpressed in cancer clusters, while *PNMT* and *PP1R1B* are only expressed in the cluster with a higher degree of immune infiltration in patient E.

Review Figure 3 | Normalized expression of ERBB2, GRB7, PNMT and PP1R1B in all clusters of Patient E.

Similar trends are observed for patient A, in the cancer in situ cluster, which is surrounded by immune cells.

Review Figure 4 | Normalized expression of ERBB2, GRB7, PNMT and PP1R1B in all clusters of Patient A.

As well as the cancer in-situ region in patient G, which just like for patient A is surrounded by immune cells.

Review Figure 5 | Normalized expression of ERBB2, GRB7, PNMT and PP1R1B in all clusters of Patient G.

PNMT is an enzyme that converts norepinephrine to epinephrine, while PPP1R1B (also known as DARP32) is usually expressed by neurons and its function is a substrate for the cAMP-dependent protein kinase (PKA), to which it may also act as an inhibitor when properly phosphorylated. [3,4] It is unlikely that the amplification of the HER2 gene and surrounding genes differ, indicating that PNMT and PP1R1B genes are somehow suppressed

in the cancer clusters. None of the genes seem native to the breast tissue, and we thus *speculate* that potentially, increased immune infiltration is observed in tumors that express such proteins as these tumors would seem more “foreign” to the immune system, than tumors that managed to somehow suppress their overexpression. This explanation is of course highly speculative, but our analysis does show some intrinsic differences between the foci.

[1] : <https://www.nature.com/articles/labinvest200819>

[2] : <https://www.ncbi.nlm.nih.gov/pmc/articles/PMC5528495/>

[3] : <https://www.genecards.org/cgi-bin/carddisp.pl?gene=PNMT>

[4] : <https://www.frontiersin.org/articles/10.3389/fnbeh.2011.00056/full>

Albeit a lack of a definitive explanation as to why this heterogeneity arises, we see it fit to bring up our observations w.r.t. heterogeneity among and within patients in the manuscript, as this showcases that such differences can indeed be seen. By establishing that ST allows for a study of heterogeneity at different levels (inter-and inpatient), we envision that our study might spark interest among other researchers and serve as a basis for more specific studies, designed specifically to survey tumor heterogeneity.

R1 | Major Comment 4

Comment : *The TLS-signature indicates the co-occurrence of B and T cells in a spot. Since a ST spot corresponds to a small region of a tumor, what is the application of the TLS-signature to bulk RNAseq profiling of tumors.*

Answer : The reviewer poses an interesting question, especially considering how much of the public data from tumor specimens that stems from bulk RNA-seq experiments rather than single cell or spatial ones. We would like to refer to our answer given to Major Question 9 Reviewer 3, but in brief, we used our TLS-signature to reproduce published data that linked a higher TLS score in bulk tumor data to increased patient survival. This analysis further supports that our TLS signature successfully stratifies patients based on their TLS content. We hope that these results should suffice as a proof of concept of how our TLS signature can be applied to bulk data, which could be extended to other datasets and diseases in future studies. The full response to question 9 reviewer 3 can be found here:

“As we’ve emphasized in other comments, we - unfortunately - are not in possession of clinical outcome data associated with our TLS-signature. However, we hypothesized that our TLS-signature is generalizable, and should not only pertain to breast cancer samples, given that TLS are also a feature of various cancer types as well as other diseases. Thus, we reasoned that the TLS model should not only be able to predict spatial location of TLS-sites in non-breast cancer tissue (as we show in Supplementary Figure 5), but also be able to reproduce results from bulk transcriptome data- regardless of the tissue - where presence/absence of TLS-signatures or alike have been associated with clinical outcome.

The publication “Tertiary lymphoid structures improve immunotherapy and survival in melanoma” (Nature 2020, <https://doi.org/10.1038/s41586-019-1914-8>), presents a study where enrichment of a TLS-signature is related to survival in the TCGA metastases dataset (Figure 3B, referenced publication). They showed how

different patient strata, when separated into three sets (low, intermediate and high) based on the TLS-signature, exhibited different survival trends (with higher TLS-score corresponding to an increased overall survival). Applying our model to predict a TLS-score for each patient, based on their bulk gene expression (downloaded from TCGA), followed by a similar stratification allowed us to successfully reproduce these results, see Figure Review Figure 20. Our findings highlight that: (1) our model agrees and produces similar results to existing - established - signatures, and (2) it is of clinical significance.

The complete analysis can be found in the annotated Rmarkdown file `tcga-analysis.R`, also provided as a compiled (knitted) pdf-file `tcga-analysis.pdf` in the github repo. Furthermore, as this analysis is now included in the main manuscript, we give a brief description of it in the Methods section and a more elaborate account in the Supplementary. As can be deduced from our replies to other comments, we have also validated the presence of TLSs in our spatial data experimentally, to add additional support to the model's and hence signature's validity. "

R1 | Minor Comment 1

Comment : *Some aspects of the results could be moved into the Methods section – for e.g. “Manual annotation”, “Initial data characterization”*

Answer : Several reviewers raised similar requests of text rearrangement, which we have tried to address. We agree that the Manual annotation and initial data characterization section were unnecessarily detailed and have chosen to join these two sections together into one more compact section, referencing the Methods for more details. We also address some of the questions posed by the reviewers here (e.g., eventual use of staining in the manual annotation and the character of the 2D UMAP-plot). The updated joint section now reads:

Manual annotation and initial data characterization

“One section from each tumor was examined and annotated by a pathologist (A.E.) based on the morphology of the associated HE-image (Hematoxylin and Eosin). Regions were labeled as either: in situ cancer, invasive cancer, adipose tissue, immune infiltrate, or connective tissue (Figure 2A and Supplementary Figure 2).

To explore the spatial gene expression data, we first applied common techniques of normalization, and visualized it in 2D-space using UMAP (Methods and Supplementary Figure 3). Spatial capture locations (hereafter: spots) from different patients separated into isolated clusters, with a high degree of intermixing between each patient's replicates. This clustering pattern implies presence of interpatient heterogeneity, as can be expected when working with tumor samples. In scRNAseq data, immune cells, but not tumor cells, typically mix between patients. In ST, multiple, often different, cell types contribute to each spot's transcription profile (~0-200 cells/spot); therefore, even immune-rich spots tend to separate in a patient-specific manner.³⁴ Thus, to properly capture the nuances of each patient's molecular profile and not risk quenching weak signals, we initially analyzed each patient separately.”

As we see it, there are two main reasons for keeping parts of these section in the main text: (i) we consider it necessary to introduce the annotations made by the pathologist before we start to compare these with our clustering result, (ii) we are keen to motivate why we take the “first separate then joint”-approach to the analysis, rather than conducting a joint analysis immediately.

We hope that the reviewer sees the value in keeping some of the information in the main text, and that our modifications render approximately the same effect as was intended with a complete “move” of these sections.

R1 | Minor Comment 2

Comment : *Text labels on Figure 2C is hard to read.*

Answer : We have increased the readability of Figure 2’s legends. Thank you for bringing this to our attention.

Reviewer 2

The paper entitled Spatial Deconvolution of HER2-positive Breast Tumors Reveals Novel Intercellular Relationships contains solid data on spatial transcriptomics of Her2+ breast cancers. It makes a first important step to better understand the organization and cell interaction in tumors and which will be very valuable for the community. However, on the formal side, I believe the authors could have added more details to allow the reader to understand and better navigate in the data. The study is decidedly interesting and clearly provides the first Spatial data in breast cancer sections but often not so well explained. It felt like it makes the reader dig into the material to understand it.

Another important formal hurdle is that the authors use single cell RNA-seq data from an unpublished paper to deconvolve their spots and see which cell types are found in each spot. This important half of the data is missing.

R2 | Major Comment 1

Comment : *The introduction is superficial and ill-suited to the content of the paper. It stresses on the clinical need for Her2 patients, but one could say that the authors have chosen this subtype because we understand to higher degree its origin and treatment: it is characterized by single major driver (the HER2 receptor), which is easily visualized and in many cases successfully targeted by antibody (trastuzumab and like), which for many works well. It is the primary and acquired resistance that render studies like the one presented here. In addition, the introduction could have engaged the curiosity of the reader by summarizing what we know about the types of cells in a breast tumor from scRNA seq, which is already a lot, to motivate on why it is important to also know where these cells are in space.*

Answer: The reviewer raised several important points in this comment, especially how our introduction had more of a clinical focus while this was not the main theme in the actual study. We have rewritten the introduction **substantially**, the whole section is hence indicated in blue and we refer to the main text rather than including a full copy of the introduction here. To summarize our changes:

- We have shifted focus from a clinical one, to rather emphasize how spatial transcriptomics studies can complement the already existing research where scRNA-seq have been leveraged to study complex tissues like breast cancer tumors. We here address the *importance* of knowing the cells' spatial locations.
- We explain why Spatial Transcriptomics (ST) is especially fit for studying cancer samples, and why it is to prefer over targeted approaches in exploratory studies like ours.
- We give a brief motivation to our interest in HER2-positive samples and why ST is a suitable technique to study these samples. We also elaborate on why characterizing the immune population and their interactions are of importance when studying the disease.

We hope that the reviewer finds that these major changes to the introduction have made the content better suited for the remaining part of the manuscript and better sparks the reader's curiosity. .

R2 | Major Comment 2

Comment : *It is unclear whether the initial Her2 staining has been used by the pathologist to identify the cancer cells or not. Or if it used at any moment in the manuscript for example when two cancer ‘clones’ are identified at for a patient. Are these two equally Her2 positive?*

Answer : We thank the reviewer for this comment and apologize for the lack of clarity. The pathologist did not use the HER2 staining to identify cancer cells, nor was it used in any other part of the analysis presented in the manuscript. We have now performed the following edits in the main text and methods

Updated Main text:

“One section from each tumor was examined and annotated by a pathologist (A.E.) based on the morphology of the associated HE-image (Hematoxylin and Eosin). Regions were labeled as either: in situ cancer, invasive cancer, adipose tissue, immune infiltrate, or connective tissue (Figure 2A and Supplementary Figure 2).

To explore the spatial gene expression data, we first applied common techniques of normalization, and visualized it in 2D-space using UMAP (Methods and Supplementary Figure 3)“

Updated Methods:

“After surgery, the tumors used for this study were trimmed of fat and divided up in several pieces, immediately frozen in -80oC, and then stored in a tumor bank until the start of the experiment. For each tumor, sections from a different piece of the tumor — to that used in the ST experiments — was subjected to IHC and PAM50 analysis for classification of subtypes. All analyzed sections from the tumors chosen for this study were stained positive for HER2 and were classified as HER2 positive tumors by PAM50. The PAM50 analysis was performed prior to the initialization of this study and was only used for sample selection.”

The question posed regarding the “HER2-ness” of the patients is intriguing, and even though we cannot use staining as a reference to gauge this, we have looked at the HER2 expression within our cancer clusters. In short, we investigated how the HER2 expression compared between clusters in the same patient, and conducted a two-sided Mann-Whitney U test to assess whether expression was significantly different between the *cancer* clusters (resulting in $(N^2-N)/2$ comparisons per patient, N = number of cancer clusters in the patient). The results can be found in Supplementary Figure 7 (also shown below) and Supplementary Table 2.

Review Figure 6 | Copy of Supplementary Figure 7, Normalized, within each patient, expression of ERBB2 (encoding the HER2-receptor). Each box represents the aggregated set of expression values for a given cluster taken over all replicates from the same patient. As can be seen, patient A exhibits high levels of ERBB2 expression in all clusters. This can be explained by the fact that the tissue samples (from patient A) have a high degree of invasive and in situ cancer, hence cancer cells are present — although often in conjunction with other cell types — in most parts of the tissue. The cancer cells constitute a “cancer background” signal that is accounted for in the contrastive differential expression analysis but emerges when looking at individual transcripts like ERBB2.

From this analysis we noted how, among certain patients, there was a statistically significant difference between the HER2-expression, implying presence of inpatient heterogeneity in HER2 expression between

different foci. The paragraph below has been added to the main text (section : Exploring Intra-and interpatient heterogeneity) :

“Given the HER2-receptor’s prominent role in the aetiology of the HER2-subtype, we also examined the ERBB2 (encoding the HER2-receptor) expression across the cancer clusters. We observed significant differences of ERBB2 gene expression (two sided Mann-Whitney U Test, $p_{adj} < 0.05$) between clusters in the same patient, attesting to a certain spatial heterogeneity of ERBB2 expression (Supplementary Figure 7 and Supplementary Table 2).”

We have also updated Figure 1, to avoid any confusion regarding what elements that were part of our study or not. See updated version below:

Review Figure 7 | Updated Main Figure 1, After sample retrieval, we performed ST (Spatial Transcriptomics) on 36 sections confirmed to be HER2-positive. A trained pathologist manually annotated one section from each sample. Expression-based clustering and single cell data integration was applied to explore the spatial expression profiles and cell type interactions in our data. Marker genes were extracted for each of the clusters and subjected to functional enrichment analysis, which allowed us to biologically annotate them. By deconvolving the expression profiles in each spot with the single cell data, we could infer patterns of cell state colocalization and design a model for prediction of TL-like structures. The single cell data and its associated cell annotations originate from an external source also examining HER2-positive tumor samples.

R2 | Major Comment 3

Comment : Additional validation based on IHC could be very valuable for example for the IFN section.

Answer: We agree with the reviewer that additional validation would be valuable and a protein based validation such as IHC is perfectly suited to validate our TL-like signature. Thus, we used dual staining of CD3 and CD20 on patient H in order to validate and visualize TL-like structures which we have included in Supplementary Figure 24 included as Review Figure 8 below.

Review Figure 8 | Reproduction of Supplementary Figure 24 ,validation of TLS-like signatures A. B and T-cell colocalization signal, derived from the single cell mapping, overlaid on the HE-image corresponding to the deconvolved ST-data. Each area containing elevated colocalization signals is enclosed by a dashed black line. B. Dual IHC staining with the T-cell marker CD3 (Vector Red (VR)/pink) and B-cell marker CD20 (DAB/brown), performed on a different section from the same tissue specimen, with the equivalent histological regions marked by a dashed line. C. 20X magnification of areas marked in B, TL-like structures are indicated by arrows D. TL-like structure from the dual-stained section of CD3 and CD20. E. The same TL-like structure from a consecutive section with respect to the dual-stain section in B, exclusively stained with CD20. F. The same TL-like structure from a consecutive sectioned to the dual-stain section in B with a single stain of CD3.

However, in order to validate the IFN signal, IHC may not be the most optimal choice since the type I interferon signaling pathway consists of multiple targets that we would need to stain against in order to confirm its presence. We considered staining for signal transducer and activator of transcription (STAT) 1 and 2 proteins which are some of the protein mediators in type-I IFN signalling, however these are not exclusive for type-I IFN but also type-III (<https://doi.org/10.4161/jkst.23931>) which made us reconsider this option. Furthermore, the STAT proteins would not have been sufficient to validate the IFN type-I signal. Upon looking into all cytokines involved in the type-1 IFN signalling pathway we found that several of them were shared with type-II IFN signalling which made us feel that we need to target a lot of them in order to really validate the correct pathway. Finally, to confirm the presence of the cell types (Macrophages and T-cells) we would have had to stain against specific marker genes of these types as well. Instead we explored the possibility to use *in situ* sequencing by Cartana (<https://www.cartana.se/>) which would have allowed us to target several components of the IFN signalling pathway (in addition to markers for T-cell and macrophage subsets) spatially and

simultaneously in a single section, we were however informed that our tissues could be treated at earliest of Q2 of 2021 and the experiment would be prohibitively expensive.

Since experimental validation of our own samples wasn't considered a feasible option, we sought to validate our findings in other data sets, as this would confirm that : (1) our observations represent more of a general mechanism and not a single isolated event, and (2) they are not artifacts from our data and the way we process it. For this purpose we used two additional data sets, the HER2-positive breast cancer sample already presented in the TLS analysis and a SCC (squamous skin carcinoma) data set. In the former we used the same single cell data as in our main study (from Wu. et al.), but in the latter we used the associated manuscript's single cell and spatial data. In both instances we **were able to reproduce the tripartite interaction** between macrophages, T-cells and a type I interferon response. We have updated the main text and added a thorough discussion (Supplementary Discussion 1) outlining this analysis in its entirety, which we also include here together with the relevant Figures (placed at the bottom of the reply).

In addition, we also expanded the analysis of our own data, showing that the signal is present in several of our samples and not only confined to a single patient, also speaking in favor of a more widespread phenomenon and this not being a singular event.

We hope that our efforts pleases the reviewer and that they instill confidence in the presence of the interaction we describe, and can be seen as an alternative validation strategy to spatially map type I IFN pathway enrichment alongside macrophage and T cell presence.

Main text (section: Spatial enrichment of type I interferon responses):

"We confirmed these findings in an independent HER2-positive spatial gene expression data set, produced using the Visium platform (Supplementary Figure 22). Finally we showed presence of the same signal in a data set from a vastly different cancer form (squamous cell carcinoma) generated with two distinct platforms, ST2K and Visium (Supplementary Figure 23 and Supplementary Discussion 1).^{54,55} Our results suggest that a spatially restricted type I interferon response may relate to M ϕ -induced recruitment of particular T-cell subsets. Further investigations would be useful to establish whether these interactions are relevant to disease outcome."

Supplementary Discussion 1 | Interferon response in the presence of M ϕ and T-cell interactions

"While patient G was used in the main text to exemplify how a joint presence of certain M ϕ (M ϕ 2:CXL10) and T-cell (T-cells:IFIT1) subsets spatially aligns with an interferon response, we were keen to chart whether this was an isolated event or could be observed among more of our samples. To answer the above question, we used a joint spot-wise score (Methods) based on the cell type proportion values; the spot-wise scores were stratified by cluster identity (Supplementary Figure 19). Several clusters seemed to be enriched for this signal, for example: pBc3, pDc5, pEc1, pGc4 (the in situ cancer cluster discussed in the main text). Of note, all the four aforementioned clusters had at least one type I interferon related pathway listed among the set of enriched pathways associated with them. We also visualize the joint scores spatially in Supplementary Figure 20, where it is clear how the elevated signals align well with the spatial regions demarcating each cluster."

The presence of a type I pathways support our hypothesis that this colocalization is partially related to a type I interferon stimuli/response, but to add support to these claims we generated spatial spot-wise enrichment plots - similar to and using the same GO-term as in Figure 4E - which are show in Supplementary Figure 21. What can be deduced from these plots is that a high joint presence of the two cell type subsets very often aligns spatially with the presence of a type I interferon pathway, but that the opposite is not always true. This is to some extent expected since type I interferon signals and responses are by no means restricted to these two cell types. Furthermore, we cannot pinpoint the likeliest source of the type I interferon response inducing stimuli (e.g., IFN α or IFN β) due to low detection of the type I interferon transcripts in this dataset

To further validate our findings, we asked whether the trifold colocalization between M ϕ , T-cells and a type I interferon response could be discerned in two additional external data sets. First, we investigated the “Human Breast Cancer (Block A Section 1)” data set from 10x GenomicsTM website, subtyped as HER2-positive; we will refer to this sample as Visium10x. We mapped the single cell data set from Wu et.al used in the main text onto the Visium10x sample using stereoscope (Methods), to obtain proportion estimates for each cell type. We found that the cell states T-cell:IFIT1 and M ϕ :CXCL10 shared similar distributions (Supplementary Figure 22A). Analogous to our analysis in the main text, we highlighted regions where the two cell states (T-cells:IFIT1 and M ϕ :CXCL10) overlap by multiplying their proportion values in each spot; again, these regions aligned with spots enriched for a type I interferon response pathway (GO:0060337) (Supplementary Figure 22B). As in the main data set, the Pearson correlation between M ϕ :CXCL10 and T-cell:IFIT showed a similarly increased signal. (Supplementary Figure 22C).

These results are encouraging as they show how the same colocalization patterns between the cell states and the interferon response signal are present in a HER2-positive breast cancer data set from: (i) an external lab (i.e., not produced by any of the authors of this manuscript) and (ii) on a different platform than what we are using (Spatial Transcriptomics 1k vs Visium). To investigate whether this phenotype could be observed in another cancer type we sought to use a second pair of data sets independent from ours. Thus, we downloaded spatial transcriptomics data (Spatial Transcriptomics 2K and Visium, n = 12 sections) and single cell data collected from Squamous Cell Carcinoma (SCC) tissue samples.⁵⁵ We were particularly interested in this data set as the authors explicitly speak of an interaction between T-cells and M ϕ (expressing CXCL9/10/11), and also note the presence of a type I interferon response in their data, but do not link these to each other. To be noted, no M ϕ subset characterized by elevated levels of CXCL9/10/11 was explicitly defined by the authors in this data set, nor any T-cell subset with an interferon related expression profile. However, a large proportion of the M ϕ cells expressed CXCL9/10/11 (see Ji.et.al Figure 4F)⁵⁵, and an interferon enriched T-cell subset could be present but with too few members to constitute their own cluster. stereoscope was used to map the SCC single cell data onto the spatial transcriptomics data (Methods). Supplementary Figure 23A shows T-cell and M ϕ cell types proportion values, where an overlap was observed. We performed identical analyses as for the Visium10x sample, computing the product between M ϕ and T-cells to expose regions where with elevated abundance of both types, to then compared the joint M ϕ and T-cell enrichment with the interferon related pathway enrichment of the type I interferon response related pathway (Supplementary Figure 23B). We also computed Pearson correlation (Methods) between M ϕ and all remaining cell types, where positive signals with the T-cells indicate a tendency to colocalize (Supplementary Figure 23C). Colocalization between T-cells and M ϕ , of varying strength, were present among all patients in the data set (Supplementary Figure 23D).

This second analysis does not only show reproducibility with a different spatial platform and a different experimental batch, but also across disease, which we consider as strong support for a mechanism involving certain T-cell and Mø states together with a type I interferon response. It also confirms that the signal is not an artifact of the single cell data set that we are using in the main analysis. However, we can yet only speculate about the mechanism of this interaction, and further - more specific and targeted - studies are required to delineate this."

Supplementary Figures associated with Supplementary Discussion 1:

Review Figure 9 | Copy of Supplementary Figure 19, Joint M ϕ and T-cell subset scores across samples and clusters, Distribution of joint proportion scores — of M ϕ 2: CXCL10 and T-cells: IFIT1 subsets — for each cluster and every patient sample. The plots are similar to violin plots, but also indicate each individual measure with a horizontal black line, medians are given by a red circle. Colors are not shared between patients.

Review Figure 20 | Copy of Supplementary Figure 20, Spatial distribution of joint Mø2 : CXCL10 and T-cell : IFIT1 scores, Distribution of joint proportion scores — of Mø2: CXCL10 and T-cells: IFIT1 subsets — for each cluster and every patient sample. The plots are similar to violin plots, but also indicate each individual measure with a horizontal black line, medians are given by a red circle. Colors are not shared between patients.

Review Figure 11| Copy of Supplementary Figure 21, Spotwise enrichment of type I interferon signaling pathway (GO:0060337) , Similarly to previous spatial plots, spots are indicated by circular markers. Cluster identities are given by the edgecolor, these colors are shared with Supplementary Figure 19. The facecolor intensity of each marker is proportional to the normalized joint scores, darker values represent high scores.

Review Figure 12 | Copy of Supplementary Figure 22, T-cell and Mø interferon response mechanism in external HER2-positive breast cancer data, A. Proportion values of T-cell:IFIT1 and Mø2:CXCL10 cell states, estimated by stereoscope. Values are scaled internally within each cell type (all values are divided by the max-value) for visualization purposes. B. (left) Spot-wise product between T-cell:IFIT1 and Mø2:CXCL10 cell states, (middle) spot-wise enrichment of the pathway “type I interferon signaling pathway” (GO:0060337), (right) combined joint product values and enrichment score; blue channel represents the joint product values and red channel the enrichment score. C. Pearson correlation between Mø2:CXCL10 and all other cell types at the minor tier, the black arrow indicates T-cell:IFIT1.

Review Figure 13 | Copy of Supplementary Figure 23, T-cell and Mø interferon response mechanism in external SCC data, A-C are equivalent to Supplementary Figure 22A-C but using the SCC single cell and spatial data, with T-cell and Mø cell types rather than finer cell states. D. Pearson correlation values estimated for the two Visium (Visium X) specimens and the four ST2K specimens (T2XY), each specimen having two replicates.

R2 | Major Comment 4

Comment : *It is unclear how the different sections from the same patient are used through the Manuscript? It seems that the sections end up showing very similar results? If true one needs a discussion on the heterogeneity and the analysis of different sections.*

Answer : The reviewer is correct in his/her observations that sections taken from the same patient are highly similar. We aim to address this point below:

All tissue sections from a patient were taken from the same specimen, with a separation of 32um or 0um (0um = adjacent section), the former applying to patient A-D and the latter to patient E-H, see Supplementary Figure 1 for reference (also included as Review Figure 14).

Review Figure 14 | Copy of Supplementary Figure 1, with the figure text : *“From patient A-D, six cryosections were collected, separated by a distance of 32 µm. From patient E-H, three consecutive cryosections were collected. Each section was taken with a thickness of 16 um and placed on an ST-array.”*

The tissue sections themselves are 16µm thick. Thus, the thickness of a section and the distance between sections are together less than the distance between the spots (200µm center to center distance). Additionally, for most of the cluster marker genes we observe a smooth (gradual increase/decrease without sharp edges) gene expression pattern, see Review Figure 15 below. Therefore, assuming that this smoothness is invariant to direction, gene expression, and thus also clustering, it is expected to be fairly similar between neighboring sections. If the reviewer is interested in an extended discussion regarding the heterogeneity between the different sections we would refer to Major Comment 1 Reviewer 1. We hope that these answers clarified the reviewers question.

Review Figure 15 | Top 3 Marker genes from each cluster in patient G visualized with section G3

Smoothness along spatial axes has also been shown in previous publications, with one example being “*Molecular atlas of the adult mouse brain*” (doi: 10.1126/sciadv.abb3446), where the molecular profile of the adult mouse brain was reconstructed in 3D using Spatial Transcriptomics data. From their results (see video

<https://www.molecularatlas.org/>) it's evident how clusters and gene expression form continuums in space rather than discrete patterns confined to each respective region.

We made some minor edits to our text to make it more explicit regarding the relationship between our sections, as this was not fully clear; more precisely, the following section has been added to the beginning of the results:

“HER2-positive tumors from eight individuals (patient A-H) were subjected to Spatial Transcriptomics (ST) with three (adjacent) alternatively six (evenly spaced) sections obtained from each tumor (n = 36 sections) (Methods and Supplementary Figure 1). For brevity, we will refer to sections originating from the same individual as replicates. Figure 1 provides an overview of the analysis workflow and methods.”

R2 | Major Comment 5

Comment : *It is almost surprising how clearly the patients are separating in the UMAP Supp Fig2. While I would clearly understand that the spots containing malignant cells may cluster by patients, I would expect that spots containing mainly immune cell would cluster together, in a patient independent way. Have the authors explored different normalization methods? Because I believe they would gain additional information by performing merged spot analysis instead of patient specific.*

Answer : The reviewer brings up an important point and we agree that a major source of heterogeneity between patient samples originates from malignant cells, but we also expect cell type composition to be a major contributor. Each spot contains a mixed transcription profile, typically originating from several cells (likely up to ~200 cells in cell dense areas). Even for spots covering non malignant cells, the diversity is rarely limited to one single cell type. With this in mind, spots would need to have nearly identical cell type composition to end up close together in the UMAP embedding under the assumption that spot expression profiles are not confounded by technical variability. Indeed, if we apply an integration method such as Harmony or CCA (Seurat) we can force the algorithm to find intersecting regions across patient datasets, but there is no guarantee that the cell type composition in these regions are identical. We did try the CCA-based integration approach on our 8 patient datasets and found similarities in gene expression profiles in spots covering non-cancerous regions such as breast glands, adipose tissues and immune infiltrates. However, such integration approaches have been developed for scRNA-seq data to group cell types of the same origin across batches that are otherwise obscured by technical variability or perturbation effects. Considering that spot expression profiles are convoluted, this assumption no longer holds and we would risk removing relevant sources of biological information which is unique to each patient sample. Regarding normalization, we have explored three different methods (1) CPM transformation, (2) log-normalization (Seurat v2) and (3) variance stabilizing transformation (VST) (Seurat v3). Just like scRNA-seq, Spatial Transcriptomics data is overdispersed[1] which led us to believe that we can benefit from using the VST which makes it possible to model overdispersion in the normalization procedure. Indeed, we found that the VST method resulted in higher performance in terms of removing sources of technical variability across spot expression profiles while retaining biologically relevant variability.

[1] : <https://www.nature.com/articles/s42003-020-01247-y>

R2 | Major Comment 6

Comment : *How many cells are captured by spot in average? Can the clustering show doublets? Multiplets? I will come back to that down with the deconvolution.*

Answer : To directly answer the reviewer; while the cell density of course is dependent on the tissue, estimates are : about 0-200 cells per spot in the ST (Spatial Transcriptomics)[1] platform, and between 1-10 cells in the Visium platform. Not operating on a single-cell level, the notion of doublets or multiplets is not fully applicable, or rather all our data points represent multiplets.

To clarify this, we have updated the manuscript to contain the following sentence to the beginning of the results section:

“In ST, multiple, often different, cell types contribute to each spot’s transcription profile (~0-200 cells/spot); therefore, even immune-rich spots tend to separate in a patient-specific manner.³⁴ Thus, to properly capture the nuances of each patient’s molecular profile and not risk quenching weak signals, we initially analyzed each patient separately.”

[1]: <https://academic.oup.com/jmcb/article/12/11/906/5861536>

R2 | Major Comment 7

Comment : *As the authors managed to deconvolve which cell types are present in each spot, is it possible to infer which cell type often ‘dominates’ spots when present in a mixture?*

Answer : The reviewer raises an interesting question; if one defines dominance as the cell type with highest proportion value in a spot, it is indeed possible to infer the “dominance” of certain cell types from our deconvolution.

stereoscope provides a “soft” classification of the spots, meaning that each spot will have Z (Z = number of cell types) values associated with it, representing the proportion values of each cell type in said spot. To find the dominant cell type, one would simply have to select the cell type with highest proportion (p_{\max}) value in each spot. Similar to what is done in the original Slide-seq publication[1] a threshold could be used in this process, such that $p_{\max} > \text{threshold}$ needs to hold in order to call a cell type dominant; here the threshold would dictate how “strong” the dominance needs to be for the cell type to be considered unambiguously dominant.

Finding dominant cell types are thus, bioinformatically, a fairly simple task once the proportion values are given. Since we provide the *stereoscope* output in our Supplementary Data 7 as well as in the github repo dominance information can easily be extracted. Thus, as we operate near the word limit already, and do not see an obvious application of this information to support or strengthen our claims we will not include it in the manuscript.

[1] : doi: 10.1126/science.aaw1219

R2 | Major Comment 8

Comment : *It is unclear if stereoscope uses the single cell data to find which single cell RNA-seq cluster / cell type map to a spot or if it searches which different cell types may be found in each spot? I find the second more interesting, the first would merely be an extrapolation on what cell type dominate the spot?*

Answer : We realize that our description of stereoscope might not have been extensive enough and have now updated the text to amend this.

Update (section “Inference of cell type organization by integration with single cell data”):

“To deconvolve our ST-data, we employed the stereoscope method, which for every spot in the spatial data estimates the proportion of cells that belongs to each cell type defined in a given single cell data set, producing a $n_{\text{spots}} \times n_{\text{cell types}}$ matrix of proportion values.”

In our answer to the specific question asked by the reviewer we partly refer to its main publication[1] for more details, but will of course also address it here in text. The output obtained from *stereoscope* is, for each spot, estimates of the proportion of cells that belong to each cell type. In other words, *stereoscope* does not simply find the most likely cell type or cluster for a spot, but decomposes the expression into contributions from each cell type. As the main text now states, the end product is a matrix $n_{\text{spots}} \times n_{\text{types}}$ where each element represents the proportion value of a given cell type in a specific spot.

[1]: <https://doi.org/10.1038/s42003-020-01247-y>

R2 | Major Comment 9

Comment : *I believe the authors developed stereoscope because ST-clusters seemed general when merging all patients from the beginning. Would it be possible to identify more specific clusters when merging data from patient-specific cluster with similar annotation?*

Answer : *stereoscope* was developed independently of this manuscript as a method to do guided deconvolution of the “mixed” ST-data (see initial bioRxiv manuscript <https://doi.org/10.1101/2019.12.13.874495>). The idea of “mapping” well annotated single-cell data spatially has gained a fair amount of traction in the field, as it is of broad interest to the community to infer cell type distributions from spatial data that lacks single cell resolution. In addition to *stereoscope*, several other approaches have either been published or uploaded to bioRxiv; examples of these are Tangram, SPOTlight, RCTD, and Seurat (part of the analysis suite, see https://satijalab.org/seurat/v3.2/spatial_vignette.html).

The method, *stereoscope*, has now been published as a stand-alone methods paper (<https://doi.org/10.1038/s42003-020-01247-y>) which we believe should: (i) validate our use of it, (ii) provide a resource where specific details can be found, and (iii) show how *stereoscope* relates to other methods designed for similar purposes.

As for the question posed by the reviewer, we must admit that we do not fully understand if the question is whether *stereoscope* would be able to find more clusters (to which the answer is that *stereoscope* does not identify new clusters, but maps those in the single cell data), or if it pertains to the expression-based clustering prior to the single cell mapping. If it is the latter that is asked; we interpret the question as asking whether similar clusters from an initial clustering analysis could be merged and then clustered again to render more specific clusters. This approach would indubitably render more clusters, but as our observations (spots) represent a **mixture** of cells and not specific cell types, our clusters are not equivalent to cell types. The interpretation of finer clusters would therefore be hard, which is why we operate at the resolution presented in the paper, which corresponds well to the pathologists annotation.

R2 | Major Comment 10

Comment : *The data used for the deconvolution of spots are not available and not even described yet, this a major drawback for this report. Stereoscope seems interesting but (i) is poorly described, (ii) is not testable, (iii) we have no clue of the data that have been used, (iv) we can't read the paper describing these data.*

Answer : As we understand it, the reviewer did not find the material we provided sufficient to assess our findings with the rigour expected and we apologize for any inconvenience. We are therefore delighted to inform the reviewer (as mentioned in earlier comments) that *stereoscope* has been published as a stand-alone method in (Nature) Communications Biology. Hopefully, this lifts some of the burden from the reviewer, given how the method now has passed the peer-review process.

Regarding the single cell data used for the *stereoscope* analysis in this study, we believe these issues are resolved; all reviewers have been given access to the single cell data via Broad's Single Cell portal as well as a (confidential) version of the manuscript to which it is associated. This is an updated version of the previously shared manuscript.

We hope that the reviewer now feels as if sufficient information is available to assess *stereoscope* and the underlying single cell dataset used for this manuscript. If any concerns remain regarding transparency or access to material, we are keen to answer and act upon these.

R2 | Major Comment 11

Comment : *It is surprising that there is so few cell types on the y-axis of Figure 3A and 3B while in the supplementary Figures there are much more cell types inferred from single cell data which seems to make much more sense*

Answer : This confusion may stem from a lack of clarity from our side. To elaborate, the subfigures 3A and 3B depict cell types from the major tier (n = 8), the referenced plots in the Supplementary illustrate the cell types in either of the finer tiers. In other words, the cell types are hierarchically ordered, first the major tier is split into several more cell types to form the minor tier, whereafter each of these cell types are split forming the subset tier. Therefore, we argue that it is not surprising to see more cell types and states (as we show in Supplementary Figure 6 of the original manuscript and Review Figure 19) compared to what we show in the main figure (major tier).

We have not updated the figure text, as this already states that the major tier is shown, but the section in the main text where the different tiers are described has been revised to add clarity.

R2 | Major Comment 12

Comment : *If stereoscope assigns the cell type dominating a spot and not all cell types found in a spot, knowing that there are several cells in a spot, would it be erroneous to seek which spots are found closer together (Fig3C) than what is expected by chance, would one need to search for which cells are more often found in the same spot than what expected by chance? The statistical analysis does not come through.*

Answer : As described in Major Comment 8 (Reviewer 2) *stereoscope* does not assign a dominant cell type to each spot but rather estimates the proportion of cells that belong to each cell type at respective spot. Had we only assigned the dominant cell type to each spot, the suggested clustering/dispersion analysis (assessing whether spots clustered more or less than what is expected by chance) analysis would have been of great interest, but is not really applicable to our data.

Instead, we use the Pearson correlation between each pair of cell types, based on their estimated proportion values in order to investigate which cell types that share similar (positive Pearson correlation) or dissimilar (negative Pearson correlation) spatial distributions. We have now updated our supplementary to describe this more thoroughly:

“We use spotwise Pearson correlation between the estimated cell type proportion values as a proxy for cell type colocalization; with high positive correlation being indicative of types that exhibit similar spatial distributions and the opposite being true for negative values. The Pearson correlations were computed across all spots for each pair of cell types; more specifically, if we let p_{sz} denote the proportion of cell type z at spot s the Pearson correlation r_{uv} between two cell types u and v would be given as:

$$r_{uv} = \frac{\sum_{s \in S} (p_{su} - \bar{p}_u)(p_{sv} - \bar{p}_v)}{\sqrt{\sum_{s \in S} (p_{su} - \bar{p}_u)^2} \sqrt{\sum_{s \in S} (p_{sv} - \bar{p}_v)^2}}$$

Where S is the set of all spots and \bar{p}_i represent the mean proportion value across all spots for cell type i , that is:

$$\bar{p}_i = \frac{1}{|S|} \sum_{s \in S} p_{si}$$

“

We hope that response made our analysis clearer, we have added the following text to the section Cell type co-localization (Methods)

R2 | Major Comment 13

Comment : *Can the data be used to understand if local inhibitory signals in the tumor do not allow B cell to differentiate in plasma cells?*

Answer : We agree with the reviewer that it would be interesting to spatially explore B-cell differentiation into antibody secreting cells (ASCs). ASCs include rapidly generated short-lived plasmablasts and terminally differentiated non-dividing plasma cells. (For simplicity’s sake, we refer to ASCs in our study as ‘plasma cells’ but it is not possible to distinguish them in our data from more short-lived plasmablasts). Whether B-cells differentiate into ASCs mainly depend on: a) initial antigen activation, b) antigen affinity, c) B cell type, and d) the type of co-stimuli.[1]. There are diverse stimuli that drive ASC differentiation; for example, toll-like receptor ligands can promote T cell independent activation, whereas CD40-ligation and cytokines are potent T-cell derived signals. Inhibitory cues that prevent differentiation to plasma cells are less characterised.

Unfortunately, we do not think our data allows us to interrogate this process in a satisfactory manner. First, it is not possible to provide information about the identity or affinity of B-cell antigens using Spatial Transcriptomics data. Second, the B-cell antigen encounter may occur in the local tumor microenvironment, e.g., in tertiary lymphoid structures, but it frequently takes place in secondary lymphoid organs, such as lymph nodes. Third, the data contains a snapshot of the tissue’s transcriptional state at the time of biopsy, which could be long past the initial antigen encounter, and it lacks single cell resolution. It is therefore very unlikely that the data is able to temporally capture B-cell antigen activation at sufficient resolution. Fourth, we want to clarify that although we find that B-cells and plasma cells occur in distinct tissue locations, our data does not necessarily suggest that

the tumor-associated B-cells do not or cannot develop into plasma cells. Additionally, the data cannot be used to determine whether the tumor associated plasma cells derive from local B-cells or not. It could simply mean that the plasma cells, post-differentiation, migrated away from where B-cells are or that the tumor-associated plasma cells differentiated from B cells at other locations outside the tumor.

Spatial Transcriptomics data is likely more suited to explore the cytokine milieu that could contribute to cellular niches, such as the plasma cell niche, since these may be continuously expressed. Several cytokines, e.g. TNFSF13 (APRIL), TNFSF13B (BAFF), and IL6) are important for plasma cell maintenance in the bone marrow.[1] Understanding if and where these stimuli are expressed could help clarify plasma cell niches outside the bone marrow. Unfortunately, these cytokines are very sparsely expressed in our dataset. TNFSF13 and IL6 were so sparsely expressed that they did not pass our initial filtration step, and as can be seen in Review Figure 16 and Review Figure 17 the expression of TNFSF13B is very low and sparse as well:

Review Figure 16 | TNFSF13 gene expression visualized on all 3 replicates from patient H

Review Figure 17 | TNFSF13 gene expression visualized on three replicates from patient G

We are therefore unable to draw any further conclusions regarding the B to plasma cell differentiation in these samples, but find the question very interesting. Future pre-clinical studies would be more apt to answer these types of questions.

[1] : doi: [10.1038/nri3795](https://doi.org/10.1038/nri3795)

R2 | Major Comment 14

Comment : *The work on TLS is interesting and shows how powerful the ST can be, at his stage the authors mention they deconvolve the cell type composition of each spot, which I guess recall one of my previous comment above. if I understand it correctly the authors should be able to give some data on how many cell types are found in each spot and within clusters if spots may seem to be more heterogeneous or not.*

Answer : The reviewer is indeed correct in the statement that we could assess the heterogeneity in our spots with respect to the cell type composition. We believe it is a good addition to our current study and something that will appeal to several readers' interest.

To quantitatively assess the heterogeneity in our data, we settled on the metric of entropy, as defined in the realm of information theory, i.e. it allows one to quantify the information contained in a random variable. For example, a distribution where all events are equally likely to happen (uniform) would have a high entropy, while a "peaky" distribution where a single outcome is favored above the others would have very low entropy. To see how these insights may be applied to our proportion estimates see below:

First, let z_s represent the cell type of a cell picked at random from a spot s . In our case z_s would be a random variable (r.v.) distributed according to $z_s \sim \text{Multinomial}(1, p_s)$, where p_s is the proportion vector (inferred by *stereoscope*) for the specific spot. With this probabilistic interpretation of the proportion vectors we can compute the entropy (H_s) of a spot s according to:

$$H_s = - \sum_{i=1}^{|Z|} p_{si} \log_2(p_{si})$$

We are thus able to measure the entropy of each individual spot. Spots with a uniform, highly mixed, cell type distribution will have a high entropy, while spots with one or a few dominant cell types have lower entropy. See Supplementary Figure 13 and Review Figure 18 for a visualization of the spotwise entropy measures.

Review Figure 18 | Copy of Supplementary Figure 13, “Distribution of entropy values within each patient and cluster. The plots are similar to violin plots, but also indicate each observation with a horizontal black line, medians are given by a red circle. Colors are not shared between patients.”

Stratifying the entropy based on the pathologist annotations, we could also see how cancer regions tended to have a lower entropy (mainly being dominated by the cancer cells) while immune infiltrates were more diverse in their character. We have also added a paragraph to the main text discussing these observations, which we’ve included below:

Spatial maps of population diversity

“To obtain an overview of the cell population’s diversity within different spatial regions, we computed the entropy for the cell type distribution (proportion estimates) within each spot (Methods). A high entropy score indicates high diversity while a low entropy score indicates the opposite. We found that the immune-related clusters were more heterogeneous than the cancer clusters, an effect more pronounced in patient A. The APC-enriched clusters, when present, tended to exhibit the highest degree of diversity, with clusters of patient B being an exception (Supplementary Figure 13).”

R2 | Major Comment 15

Comment : *It is stated that the pathologist annotation is an important first step to validate the results, i.e. to show a correspondence between the visual inspection and the expression data. While the plot in Fig 1E shows how the pathologist and ST agree, it could be interesting to understand more the disagreement, f.ex. to test whether happens at the margins or more globally.*

Answer : The extent of agreement (or disagreement) between the manual annotations from the pathologist and annotations derived from the expression-based clustering is an interesting topic, especially given the attention that “Digital Pathology” has been given in the field of Machine Learning (ML) and Computer Vision.

Still to make general claims about how expression-based clustering and annotations from a pathologist relates to each other, we would need to systematically assess large datasets that are annotated by multiple pathologists. Applying a study of this kind to our data, with a sample size of 36 and with a single pathologist, would only answer the question of how our particular clustering differs from said pathologists work.

We are still convinced that the annotations indeed work as a good framework to operate by, and we would like to emphasize that with this study we aim to define spatial cellular patterns within breast tumor samples. As it stands, we are unable to provide a robust evaluation of the benefits and drawbacks of manual versus digital annotation.

R2 | Minor Comment 1

Comment : *In the beginning of the results, some Figures are shown before being cited example Figure 2B and 2C*

Answer : We appreciate that the reviewer highlights these inconsistencies. We have done our best to make sure that the figures are shown after being cited in the text.

R2 | Minor Comment 2

Comment : *Please work on the legends of Figures, there are many clusters and colors and it is sometimes difficult to follow which cluster is mentioned and with each color. For example, legends are missing for Supp Fig.5.*

Answer : It is much appreciated that these details are brought to our attention, and we thank the reviewer for it. Figure legends have now been added to Supplementary Figure 5 (now 6) and 15 (now 25).

R2 | Minor Comment 3

Comment : *It is complicated to follow the terminology major / minor tier when one has not read the paper (in preparation) introducing this terminology.*

Answer : We apologize for any inconvenience that this may have caused during the review process. In order to clarify this point, we have attached Figure 6A from the original manuscript, it outlines the hierarchical relationship among cell types and tiers.

Review Figure 19 | Figure 6 from the Wu et al. manuscript, the source of our single cell data

Reviewer 3

Andersson et al. used single-cell mRNA data to deconvolve spatial transcriptome and identified cell type differences in different spatial regions. They reported that expression-based clustering enables data-driven tumor annotation and assessment of intra-and interpatient heterogeneity. They discovered segregated epithelial cells, interactions between B and T-cells and myeloid cells, co-localization of macrophage and T-cell subsets by using Spatial Transcriptomics technology. They also constructed a model to infer the presence of tertiary lymphoid structures, applicable across tissue types and technical platforms. According to their finding, authors bring out new tools and biological insights into the spatial organization in HER2-positive breast cancer tumors, define novel interactions between tumor-infiltrating cells, and may open up for new therapy for HER-2 positive breast cancer tumors.

It is an interesting work with novelty, which combined tumor sequencing strategies with spatial information to explore HER-2 positive breast cancer. Even though the method is innovative, and the data is reproducible, there is not much controversy in terms of bioinformatics. Authors also need to depict more common characteristics or subpopulation analysis among all eight tumor samples instead of, for example, major attention to patient G and H. Co-localization of immune cells, such as TLS, CAFs, with cancer cells could be explored as well. Clinical significance of their finding were fully presented. This study does not seem to provide sufficient detailed insights into the role of B and T-cells and myeloid cells in tumor microenvironment with compelling translational implications.

R3 | Major Comment 1

Comment : *The pathologist in Figure 2B identifies cell types in different areas. But is there any counting of the number and proportion of different types of cells in different regions? Whether these ratios can be used as a reference for further deconvolution.*

Answer : We again agree that inclusion of cell segmentation (and implicitly counting of cells) in theory could be incorporated into the *stereoscope* model, and refer to the response given to reviewer 1 Major comment 2 for a discussion regarding this idea, included below.

Also, to directly answer the question about whether cells have been counted, the answer is no, this has not been done by either pathologists or computationally.

“Answer (Reviewer 1 Major Comment 2): *Coupling of cell segmentation to the inference process could theoretically be incorporated into the model, for example as a way to introduce a lower bound for the proportion values, to exemplify: If 10 cells are located at a spot, no proportion value should be lower than 0.1 (1 / 10) since this would represent less than one cell. The - unpublished - method Tangram (<https://www.biorxiv.org/content/10.1101/2020.08.29.272831v1>) leverage such information in their optimization-based approach (though mainly for image based Spatial Transcriptomics techniques like MERFISH and seqFISH). The method we are using (now published in *Communications Biology*), *stereoscope*, however does not currently support incorporation of cell segmentation information.*

stereoscope was specifically designed for sequencing-based methods (e.g. ST, Visium and Slide-seq), where only some platforms provide paired HE-images of the tissue. Therefore, we partially sought to develop a method that would not rely on segmentation data. Furthermore, performing cell segmentation on images of the fairly thick tissue sections used in ST and Visium, with multiple cell layers in the z-axis, is non-trivial and we are not aware of any robust methods devised for this purpose. Introduction of cell segmentation information would also

introduce a lot of uncertainty in the inference process and potentially confound the results. We have tried to apply cell segmentation software in prior studies, and noted how the aforementioned issues are very prominent in cell dense regions (which we have plenty of in our cancer data).

Nevertheless, we fully share the reviewer's opinion that inclusion of additional data (such as cell segmentation) could enhance the stereoscope results and would be a valuable addition to the method itself. When updating the stereoscope software, we will most certainly keep this suggestion in mind."

R3 | Major Comment 2

Comment : *The UMAP plot labeled with different samples in Supplementary Figure 2A shows that the samples of different patients present a discrete distribution, which indicates that the transcriptome of each patient is different. However, if one has analyzed single-cell RNAseq, will know that only cancer cells (Malignant cells) show differences between patients, so how to explain the differences between patients in this entire region contains all types of cells?*

Answer : The reviewer is fully correct in this observation, and that it to some extent differs from what one tends to see in single cell data. However, the ST-data is very different in character and composition from the single cell data, and slightly different results are to be expected. We elaborate more about this, and why this partitioning of spots into discrete subsets did not raise a serious concern for us during the analysis in our answer to Major comment 5 from Reviewer 2, which we refer to in order to avoid redundancy (see below). We have also addressed this issue in the main text, to be more explicit about the difference between scRNA-seq and ST data:

Updated Main text (section, Manual annotation and initial data characterization)

"In scRNAseq data, immune cells, but not tumor cells, typically mix between patients. In ST, multiple, often different, cell types contribute to each spot's transcription profile (~0-200 cells/spot); therefore, even immune-rich spots tend to separate in a patient-specific manner."

Major comment 5 Reviewer 2: *The reviewer brings up an important point and we agree that a major source of heterogeneity between patient samples originates from malignant cells, but we also expect cell type composition to be a major contributor. Each spot contains a mixed transcription profile, typically originating from tens of cells (likely up to ~200 in cell dense areas). Even for spots covering non malignant cells, the diversity is rarely limited to one single cell type. With this in mind, spots would need to have close to identical cell type composition to end up close together in the UMAP embedding under the assumption that spot expression profiles are not confounded by technical variability. Indeed, if we apply an integration method such as Harmony or CCA (Seurat) we can force the algorithm to find intersecting regions across patient datasets, but there is no guarantee that the cell type composition in these regions are identical. We did try the CCA-based integration approach on our 8 patient datasets and found similarities in gene expression profiles in spots covering non-cancerous regions such as breast glands, adipose tissues and immune infiltrates. However, such integration approaches have been developed for scRNA-seq data to group cell types of the same origin across batches that are otherwise obscured by technical variability or perturbation effects. Considering that spot expression profiles are convoluted, this assumption no longer holds and we would risk removing relevant sources of biological information which is unique to each patient sample. Regarding normalization, we have explored three different methods (1) CPM transformation, (2) log-normalization (Seurat v2) and (3) variance stabilizing transformation (VST) (Seurat v3). Just like scRNA-seq, Spatial Transcriptomics data is overdispersed which led us to believe that we can benefit from using the VST which makes it possible to model overdispersion in the normalization*

procedure. Indeed, we found that the VST method resulted in higher performance in terms of removing sources of technical variability across spot expression profiles while retaining biologically relevant variability.

R3 | Major Comment 3

Comment : *The author proposed the stereoscope algorithm to deconvolve the spatial transcriptome. However, the window of deconvolution seems to have only selected 1 spot. Is it possible to try to deconvolve a window of 2x2 or 4x4 spots to test the robustness of the algorithm on a larger spatial scale?*

Answer : The suggestion of aggregating multiple spots, or alternatively increasing the deconvolution area is interesting and would pose an opportunity to evaluate the stereoscope method's robustness. However, as this paper is only applying *stereoscope* (now published in Communications Biology [1]) we would like to argue that it is out of the scope of this work to validate and assess robustness of the method we are using. Of course, we understand the reviewer's request and consider it highly reasonable as *stereoscope* had not yet been published upon submission of this manuscript; but, since the method now has been through the peer-review process we feel justified in this argumentation and hope the reviewer agrees.

[1]: <https://doi.org/10.1038/s42003-020-01247-y>

R3 | Major Comment 4

Comment : *Cell interactions need to connect with clinical significance.*

Answer : We fully agree that it would be very interesting to relate our findings to clinical outcome. However, as we discuss in other replies (Major Comment 10), we are unfortunately not in possession of any such patient metadata. Also, even if we had such data - a study with 8 individuals would lack the necessary statistical power to relate cell interactions with clinical outcome.

While we cannot link all our findings to disease outcome, our study is the first of its kind: applying ST to multiple sections across several patients all diagnosed as being HER2-positive. The study was intended to be exploratory, showcasing how computational methods applied to ST data enables new insights - on a molecular level - to be gained for diseases like HER2-positive breast cancer. We also envision that many studies will follow this one, where clinical correlations and relationships are the main focus. However, we envision that the study presented here will serve as a first "stepping-stone" towards a spatially focused exploration of the disease's molecular composition.

R3 | Major Comment 5

Comment : *Validate expression of the myeloid cell gene signature and the T-cells gene expression modules in the TCGA.*

Answer : Here we must unfortunately ask for further clarifications, as we are slightly unsure of exactly which gene signature the reviewer is referring to. Our inference of the T-cell and myeloid interaction is not based on any specific signature, but rather cell type correlation values (derived from the cell type mapping using a large set of ~5000 genes).

If it is the presence of an interaction between the two cell types that we are asked to assess further using TCGA, we would like to emphasize that we are by far not the first to claim that such an interaction exists,[1-4] but merely suggesting that we have indications of its presence in our data as well. However, we appreciate the request for further validation on external data, but would argue that this interaction signal is best studied in

spatial and single cell data rather than the TCGA. Thus, to corroborate our results, we have conducted two additional analyses using external data. We present these results in Supplementary Discussion 1 and discuss them further in our reply to Major Comment 3 Reviewer 2 (due to the length of this reply we do not include a copy of it below).

We hope that these new analyses are satisfactory to the reviewer, as we now show the presence of the trifold interaction between macrophage and t-cell states with the interferon signal in a total of three (including our own) data sets.

[1] : <https://doi.org/10.1016/j.cels.2017.12.001>

[2] : <https://doi.org/10.1158/1078-0432.CCR-19-1868>

[3] : <https://doi.org/10.1016/j.ccell.2016.03.005>

[4] : <https://doi.org/10.1016/j.ccell.2019.05.004>

R3 | Major Comment 6

Comment : *More common characteristics of all eight tumor samples should be described such as cell types and co-localization information instead of, for example, major attention to patient G and H.*

Answer : We understand how putting major focus on a select set of patients like G and H might give the impression that we are neglecting certain aspects of our data, this was not our intention. In most cases, we chose these patients to visualize our findings since all samples would not fit into a single figure. Nonetheless, as a response to this and other comments, we have added a paragraph and one Supplementary Discussion associated with three Supplementary Figures (Supplementary Figures 19-21) that describes colocalization of the macrophage and T-cell subset across all samples as well as the spatial enrichment of an interferon related pathway. In addition, we have also added commentary regarding how the gene ERBB2 is expressed across patients and clusters. Finally, we also discuss more common features for the non-immune cells in a new paragraph added to the section *“Trends of cell type co-localization”*.

We would also like to stress that even though more focus is given to the two aforementioned patients, we do discuss observations from other patients throughout the whole manuscript. In the section *“Exploring intra-and interpatient heterogeneity”* patient E is given plenty of attention due to the presence of different tumor foci. Also, Patient B is discussed in more detail within the section *“Enrichment of cell types within manually defined regions”*. We further describe global features shared among all our samples in the section *“Trends of cell type co-localization”*, before narrowing down the discussion to patient G and H. Finally, by deriving and extracting the core signatures we also aimed to describe common features shared across all patients.

We hope that these additions to the already existing material satisfies the reviewer’s wish to see more broad discussion of the whole dataset.

R3 | Major Comment 7

Comment : *More cell subpopulation analysis should be presented. Besides interactions between B and T-cells and myeloid cells, author should deconvolve cell state relationships in the tumor microenvironment.*

Answer : Please see the answer to comment 8 below

R3 | Major Comment 8

Comment : *Co-localization analysis of immune cells and cancer cells could be explored instead of co-localization analysis restricted only among immune cells.*

Answer : The interactions between immune and tumor cells are indeed highly relevant, and we would like to emphasize how we do include results that account for interactions between both tumor and immune cells as well as among different types of tumor cells (see colocalization matrices at the different tiers). We provide all correlation matrices in Supplementary Data 9, which allows anyone with an interest to further explore signals different from those we focus on in the manuscript can pursue this interest. Having surveyed our data, we must however say that it is hard to see significant correlations between tumor and immune cells, most likely because cancer cells tend to be very dominant in the spots they contribute to. To keep our study within the content limit, we decided to focus on immune-immune interactions.

Still, to provide a more holistic commentary to our results we have updated the text to: (1) highlight more types of interactions than in the previous iteration; (2) be more upfront in the discussion by stating that we've by no means exhausted our data's full potential. The added paragraph is found in the section "*Trends of cell type co-localization*", and is also included below:

"The cell mapping was used to explore putative cellular interactions by computing the cell type spot-wise Pearson correlation (Figure 3C), with a positive correlation between two cell types being considered as them colocalizing. At the major tier, the most prominent feature — present in all patients — was an anticorrelation between epithelial cells and every other cell type (Figure 3C). The endothelial cells (major tier) also exhibited well-preserved colocalization patterns with CAFs (all patients except G) and perivascular cells (all patients except F)(Figure 3C). At the minor tier, epithelial cells are split into cancer and normal epithelial cells, which anticorrelate in all patients except patient E (Supplementary Figure 14). For patient-wise colocalization matrices we refer to Supplementary Data 9. The increased cell type resolution thus revealed how the cancer epithelial cells are the main contributors to the trend of epithelial cell anticorrelation observed in the major tier. There is also a spatial separation of the two CAF types at the minor tier (all patients except patient A, Supplementary Figure 14). At the subset level mature Luminal cells (a subset of normal epithelial cells) always colocalize with one cancer type (except in patient A), which could suggest proximity to luminal cells or a mature luminal phenotype (Supplementary Figure 15)."

We hope that our response together with the associated manuscript updates are found to be satisfactory.

R3 | Major Comment 9

Comment : *The connection of TLS-score which was based on its (normalized) gene expression with clinical significance like disease outcomes could be added to further verify the function of TLS identified by the model. Because the real TLS include many other immune cells besides T and B cells, and the T and B cells might be in some specific states in TLS.*

Answer : This comment, in conjunction with other expressions of a desire to relate our results to clinical outcome or implications prompted us to reassess the options for such analyses. As we have emphasized in other comments, we - unfortunately - are not in possession of clinical data associated with our patients; but we to some extent make the claim that our TLS-signature is generalizable, and should not only pertain to breast cancer samples. Thus, we reasoned that if our claims were true - the TLS model should be able to reproduce results - regardless of the tissue - where presence/absence of TLS-signatures or alike have been associated with clinical outcome.

The publication “*Tertiary lymphoid structures improve immunotherapy and survival in melanoma*” [1], presents a study where enrichment of a TLS-signature is related to survival in the TCGA metastases dataset (Figure 3B, referenced publication). They showed how different patient strata, when separated into three sets (low, intermediate and high) based on the TLS-signature, exhibited different survival trends (with higher TLS-score corresponding to an increased overall survival). Applying our model to predict a TLS-score for each patient, based on their bulk gene expression (accessed from TCGA), followed by a similar stratification allowed us to successfully reproduce these results, see Review Figure 20. Our findings highlight that: (1) our model agrees and produces similar results to existing - established - signatures, and (2) it is of clinical significance.

Our Results

Cabrita et.al Figure 3B

Review Figure 20 | Kaplan-Meier plots of trichotomized, stratified by TLS-signal strength, TCGA-melanoma data. Left panel (“Our results”) shows the result when our predictive TLS-model was applied to estimate TLS-signal, Right panel (“Cabrita et.al Figure 3B”) shows the results presented in the Cabrita et.al paper when employing their method to estimate TLS-enrichment metric.

The complete analysis can be found in the annotated Rmarkdown file `tcga-analysis.R`, also provided as a compiled (knitted) pdf-file `tcga-analysis.pdf` in the github repo. Furthermore, as this analysis is now included in the main manuscript, we give a brief description of it in the Methods section and a more elaborate account in the Supplementary. As is mentioned in our previous replies to other comments, we have also validated the presence of TLSs in our spatial data experimentally, to add additional support to the model’s and hence signature’s validity.

[1]: <https://doi.org/10.1038/s41586-019-1914-8>

R3 | Major Comment 10

Comment : *Type I interferon promoting macrophage-induced recruitment of T-cells could be further investigated its relationship with disease outcomes.*

Answer : It would be extremely interesting to connect the presence of the type I interferon pathway to clinical data, and to see whether the presence or even extent of the T-cell recruitment holds any predictive power. We are nevertheless not in possession of any clinical outcome data for these patients. While unfortunate, this to some extent shows the value of our publication; namely, how it acts as a showcase of what type of information that can be extracted by applying spatial transcriptomics to tissue sections, and inspiration for further studies to be conducted, designed to answer the questions we are not able to fully address here. To encourage and emphasize this, we have updated the Discussion to include more comments on the clinical relevance of this study.

R3 | Minor Comment 1

Comment : *There was only one pathologist to estimate the various region. Two or more pathologists work together to judge might be more reliable.*

Answer : We fully agree that two pathologists would have been ideal, as it is widely known that pathologists tend to vary in their assessment of samples. The pathologist's annotations are broad and not key to any of our analyses, but act as a framework which we compare our results to, mainly to show that the expression-based clustering "makes sense". Our confidence in the pathologist's annotation is also increased, just because of the high concordance with the data-driven expression-based clustering, as this represents convergent results from two vastly different methods (visual inspection and clustering). We have therefore not requested similar broad annotations from a second pathologist, and hope the reviewer understands our reasoning.

R3 | Minor Comment 2

Comment : *Plots in Figures should be labeled with A, B, C., for example, Supplementary Figure 12-14*

Answer : We welcome these comments and thank the reviewer for paying attention to such details. These omissions of subplot labels have been corrected and all the supplementary figures should now be formatted correctly.

Reviewer 4

R4 | Major Comment 1

Comment : *The paper is not transparent regarding the data and methods used. I do not believe that it would be acceptable to publish this manuscript before the data from the other unpublished paper is made available (ref. 30, used in throughout the manuscript). Second the authors use the stereoscope method (Figure 3, ref 22) but this has not been published. How can we evaluate the use of this method when it has not been published? For example, the method is similar in output to the Moncada et al publication (ref 21); however this is not acknowledged or compared.*

Answer : The concerns raised by the reviewer are both well-motivated and legitimate. We have done our best to address the issues brought up by the reviewer.

First, *stereoscope* has now been published in a peer-reviewed journal (Communications Biology).[1] There is also, as referenced in the previous version of the manuscript, a tutorial together with open-source code hosted on github. Second, we now reference Moncada et al's results in our manuscript to highlight that other methods for the same task exist. Unfortunately, we were unable to access the code to reproduce Moncada's results, but we hope that the addition to the text will satisfy the reviewer.

Third, we are now able to provide access to the full single cell data set used in the publication via Broad's Single Cell Portal, which should allow all reviewers to properly probe and examine this data. Details for accession are provided at the end of this document. We can also inform the reviewers that the paper from which the single cell data originates (Wu. et al.) is in the second phase of revision in a Nature journal. Combined, we believe that these additions serve to provide sufficient information about all aspects of this publication for review and apologize for any confusion or inconvenience we may have caused.

[1] : <https://doi.org/10.1038/s42003-020-01247-y>)

R4 | Major Comment 2

Comment : *In the section on spatial enrichment of type I interferon response the results are only shown for patient G. The authors should show a result that holds for more of the patients and show that in a main Figure.*

Answer : We recognize the reviewer's suggestion to show the interferon response for more than a single patient (patient G), as this would imply some extent of reproducibility and minimize the risk of our observations being due to confounding factors or merely consisting of spurious noise. We'd also like to raise a very important point in this context, being that we do not expect to see a type I interferon response in all of our patients; albeit reported in earlier studies, the type I interferon response is not known to be a universal feature of cancer in general nor for our particular subtype. That being said, to inquire whether other regions potentially exhibited signs of similar interactions we leveraged an approach similar to that used in the deduction of potential TLS-sites in our tissue sections. More specifically, we multiplied the estimated proportion values from the Macrophage 2 : CXCL10 subset with those from the T-cell : IFIT1 subset; this produced a new signal with values related to the joint presence of both types. We normalized the proportion product values within each section by a z-transformation (mean subtraction and division by standard deviation).

Next, we stratified the spots in each section with respect to cluster identity and assessed the distribution of proportion products within each cluster - these are now included in the manuscript as Supplementary Figure 19 shown below as Review Figure 21.

Review Figure 21 | Copy of Supplementary Figure 19, Joint Macrophage and T-cell subset scores across samples and clusters. Distribution of joint proportion scores — of Macrophage 2 : CXCL10 and T-cells : IFIT1 subsets — for each cluster and every sample. The plots are similar to violin plots, but also indicate each individual measure with a horizontal black line, medians are given by a red circle. Colors are not shared between patients.

As expected, cluster 4 in patient G, the cluster discussed in the main text, showed elevated values compared to the majority of the clusters. Furthermore, we saw elevated signals within a few other clusters as well, for example cluster 3 of patient B - annotated as enriched for IFN response pathways, and to which marker genes

like CXCL10, IFI6, IFIT1, IRF7 had been assigned. When visualizing the spatial enrichment of the same interferon response pathway as shown for patient G (Figure 4), we noted how these aligned well with the clusters with an elevated joint signal, implying presence of the aforementioned set of interactions. We also include spatial enrichment plots of said interferon response pathway in Supplementary Figure 21, shown for all samples.

We have updated the text to reference these figures and the section (Supplementary Discussion 1) that describes our analysis in more detail as to speak of this signal in a more global context, and not only focusing on patient G, we also extended our analysis beyond our own data, examining two external data sets which both showed affirmative results. We have however not updated our main figures, as we believe exemplifying our results with one (or at most two patients as in Figure 5) makes it easier for the reader to follow the results we present.

R4 | Major Comment 3

Comment : *The validation of the TLS signature on a single other instance in Fig. 5D is not sufficient. Also since the pathologist is part of the team I'm not sure that it can be seen as an independent annotation.*

Answer : We agree that a single instance is insufficient to validate the TLS-model used to predict the TLS-score. Therefore, we included three additional (four with the Visium data) datasets in our study, shown in Supplementary Figure 25, which we now have chosen to highlight more prominently in the manuscript.

In said figure, the model is applied to: i) melanoma samples where lymphoid areas had been marked by a pathologist, ii) rheumatoid arthritis (RA) where ectopic lymphoid structures (generally used as a synonym for TLS) also were identified by a pathologist, used as a positive control; iii) the breast cancer Visium sample shown in Figure 5D; and iv) a tissue sample from the developmental heart, where no strong signal is expected, used as a negative control. To clarify, the RA and melanoma samples were made by two different pathologists not involved in this study.

In response to this comment and that of other reviewers, we also extended the TLS analysis. First we have now validated our findings with IHC staining. Which is shown in Supplementary Figure 24 here included as Review Figure 22.

Review Figure 22 | Copy of Supplementary Figure 24, IHC validation of TLS-like signatures

A. B and T-cell colocalization signal, derived from the single cell mapping, overlaid on the HE-image corresponding to the deconvolved ST-data. Each area containing elevated colocalization signals is enclosed by a dashed black line. B. Dual IHC staining with the T-cell marker CD3 (Vector Red (VR)/pink) and B-cell marker CD20 (DAB/brown), performed on a different section from the same tissue specimen, with the equivalent histological regions marked by a dashed line. C. 20X magnification of areas marked in B, TL-like structures are indicated by arrows. D. TL-like structure from the dual-stained section of CD3 and CD20. E. The same TL-like structure from a consecutive section with respect to the dual-stain section in B, exclusively stained with CD20. F. The same TL-like structure from a consecutive section to the dual-stain section in B with a single stain of CD3.

As can be seen from Review Figure 22, we were able to identify TL-like structures in all four regions where we observed a high TLS-score in patient H, confirmed by both single and dual staining. These structures however lacked clear germinal centers, hence why we've proceeded to use the term TL-like structure rather than the more clinical "TLS" term. We also attempted to perform similar staining on new sections from patient G, but due to the structure of the tissue - now containing larger holes in the tissue - it was impossible to section thin enough sections without tearing it. Because of this we concluded that they were unfit to be used as validation, even though the areas where we observed signal from both CD3 and CD20 correlated well with the expected areas. We have included examples of our staining attempts with tissue from patient G in Review Figure 23

Review Figure 23 | IHC staining of patient G with the same setup as done for patient H, where CD3 was developed with Vector Red, CD20 developed with DAB and then hematoxylin used as counterstain. **Top left** : single stain against CD3, **bottom left** : single stain against CD20, **top right** : double stain against CD3 and CD20, **bottom right** : negative control.

We have also included the following section in the main text, as well as added a more explicit description of the procedure to the supplementary as well as made appropriate updates to the Methods section:

“Experimental validation of TL-like structures

To validate that our estimates of joint B and T-cell presence were suitable as proxies for TL-like structures, we used IHC (immunohistochemistry). We stained against the canonical cell type markers CD20 (B-cells) and CD3 (T-cells) on three additional sections from patient H, one double staining and two single stains (Methods). Our results showed characteristic features of previously described immature TLSs — compartmentalized structures of B-cells surrounded by T-cells, but lacking distinct germinal centers — in regions of elevated TLS-signals (Supplementary Figure 24).”

As additional validation, we also applied our model to bulk data and managed to produce results relating TLS-presence to clinical outcome, see Supplementary Section 3.

With this additional support for our data, we hope that the reviewer finds that more rigorous evidence for our claims now is presented.

R4 | Major Comment 4

Comment : *The statistics is generally not well documented in the manuscript. P-values should be given in the text , here for example: "but one of the cluster displayed a high degree of enrichment for immune response related processes (cluster 3, patient E) while apoptotic and regulatory pathways were enriched in the other (cluster 4, patient E), see Supplementary Data 6 ." Is the reader expected to go into the supplementary data file to find whether the result is statistically significant?*

Answer : We are grateful that the reviewer brought this lack of documentation to our attention. The reader should of course have immediate access to the p-values and not be forced to search the supplementary data. We have added statistical values and better documentation to our analysis throughout the manuscript.

R4 | Major Comment 5

Comment : *The writing in the Results section is not well-organized and contains many statements that are not fitting for it. For example, "Of course, the presence of multiple cancer clusters does not necessarily reflect different tumor types, but rather that the corresponding spatial regions are not homogenous in their expression; for example, as a consequence of hosting distinct immune cell populations. If additional metadata could be obtained for patients, ST might be useful for relating tumor intrapatient heterogeneity to more quantitative metrics, such as survival or treatment response."*

Answer : We thank the reviewer for this comment and have performed extensive editing of the manuscript to address this point. In this updated manuscript version, we have made several rearrangements of the text. Such as:

- Moved paragraphs that comment on or discuss results rather than present them from the Results to the Discussion including the paragraph referenced above (now in the Discussion).
- Removed or shortened paragraphs that were unnecessarily descriptive in favor of referencing to the Supplementary or Methods. Examples being the explanation of our spatial pathway enrichment analysis and TLS-score definition.
- Rewritten certain paragraphs and sections in the Results to make them more condensed and shorter. For example the single cell integration section.

We hope that our changes are satisfactory and that the reviewer finds the new text organization more appropriate.

REVIEWERS' COMMENTS

Reviewer #1 (Remarks to the Author):

The authors have addressed all of my questions. This is an important study and I recommend publication.

Reviewer #2 (Remarks to the Author):

The paper entitled Spatial Deconvolution of HER2-positive Breast Tumors Reveals Novel Intercellular Relationships has been substantially rewritten to meet the reviewer's comments. The comments of all three reviewers and the corresponding answers have created an interesting discussion and the result is a comprehensive study of cellularity and spatial distribution of features. The introduction has been re-written to better correspond to the content of the paper. All comments to reviewers have been treated with attention and detailed answers. I have no additional comments to this paper except to recommend its quick expediting, while the novelty is still high, as this group of authors deserve the credit.

Reviewer #3 (Remarks to the Author):

The author has answered all my questions and I suggests accepting this paper.

Reviewer #4 (Remarks to the Author):

The authors have addressed all of my concerns. I have also read through the concerns of the other reviewers and the responses of the authors. Overall, it seems to me that the paper is ready for publication at Nature Communications.